# Solid-State $^{1}$H Spin Polarimetry by $^{13}$CH$_3$ Nuclear Magnetic Resonance

Stuart J. Elliott[1,2], Quentin Stern[1] and Sami Jannin[1]

[1]Centre de Résonance Magnétique Nucléaire à Très Hauts Champs - FRE 2034 Université de Lyon / CNRS / Université Claude Bernard Lyon 1 / ENS de Lyon, 5 Rue de la Doua, 69100 Villeurbanne, France
[2]Department of Chemistry, University of Liverpool, Liverpool L69 7ZD, United Kingdom

*Correspondence to*: Stuart J. Elliott (Stuart.Elliott@liverpool.ac.uk)

**Abstract.** Dissolution-dynamic nuclear polarization is used to prepare proton polarizations approaching unity. At present, $^{1}$H polarization quantification remains fastidious due to the requirement of measuring thermal equilibrium signals. Lineshape polarimetry of solid-state nuclear magnetic resonance spectra is used to determine several useful properties regarding the spin system under investigation. In the case of highly polarized nuclear spins, such as those prepared under the conditions of dissolution-dynamic nuclear polarization experiments, the absolute polarization of a particular isotopic species within the sample may be directly inferred from the characteristics of the corresponding resonance lineshape. In situations where direct measurements of polarization are complicated by deleterious phenomena, indirect estimates of polarization using coupled heteronuclear spins prove informative. We present a simple analysis of the $^{13}$C spectral lineshape of [2-$^{13}$C]sodium acetate based on the normalized deviation of the centre of gravity of the $^{13}$C peaks, which can be used to indirectly evaluate the proton polarization of the methyl group moiety and very likely the entire sample in the case of rapid and homogeneous $^{1}$H-$^{1}$H spin diffusion. For the case of positive microwave irradiation, $^{1}$H polarization was found to increase with an increasing normalized centre of gravity deviation. These results suggest that, as a dopant, [2-$^{13}$C]sodium acetate could be used to indirectly gauge $^{1}$H polarizations in standard sample formulations, which is potentially advantageous for: (*i*) samples polarized in commercial dissolution-dynamic nuclear polarization devices that lack $^{1}$H radiofrequency hardware; (*ii*) measurements that are deleteriously influenced by radiation damping or complicated by the presence of large background signals; and (*iii*) situations where the acquisition of a thermal equilibrium spectrum is not feasible.

## 1 Introduction

Classical nuclear magnetic resonance (NMR) experiments produce inherently weak signals. The severely limiting low intrinsic sensitivity of the technique can be enhanced by up to four orders of magnitude by employing a wide range of routinely used hyperpolarization methodologies (Ardenkjær-Larsen et al., 2003; Hirsch et al., 2015; Dale and Wedge, 2016; Meier 2018; Kouřil et al, 2019). The significantly boosted NMR signal intensities from metabolites hyperpolarized by implementing a dissolution-dynamic nuclear polarization (*d*DNP) approach have been used in the clinical diagnostics of cancer in human patients (Nelson et al, 2013; Chen et al, 2020; Gallagher et al, 2020).

To hyperpolarize nuclear spins via the *d*DNP approach, the spin system of interest is co-frozen in a mixture of aqueous solvents and glassing agents with a carefully chosen paramagnetic radical species (Abragam and Goldman, 1978). The *d*DNP-compatible solution is subsequently frozen at liquid helium temperatures, where the solvent matrix forms a glass, inside a magnetic field and is irradiated with slightly non-resonant microwave irradiation, which transfers the high electron spin polarization to the nuclear spins of interest (Kundu et al, 2019).

Hyperpolarization of methyl group moieties by *d*DNP has led to some unusual effects including the generation of long-lived spin order, which is revealed in the liquid-state upon dissolution of the material from cryogenic conditions (Meier et al, 2013; Roy et al, 2015; Dumez et al, 2017; Elliott et al, 2018). Solid-state NMR of highly polarized nuclear spins has previously been utilized to infer the sample polarization level and, in suitable cases, the quantity of long-lived spin order established (Elliott et al, 2018; Waugh et al, 1987; Kuhns et al, 1989; Marohn et al, 1995; Kuzma et al, 2013; Mammoli et al, 2015; Willmering et al, 2017;

Aghelnejad et al, 2020). To the best of our knowledge, the solid-state NMR spectra of strongly polarized methyl groups have not shown any significant features which may be used for a clear lineshape analysis.

In this Communication, we propose that the $^{13}$C NMR lineshape of [2-$^{13}$C]sodium acetate can be used to indirectly quantify the $^1$H polarization of the methyl group spins. Furthermore, since $^1$H-$^1$H spin diffusion rapidly achieves a homogeneous proton polarization across the entire sample, the $^1$H polarization level of the whole sample is therefore likely to be reflected by the $^1$H polarization of the methyl group moiety. We analyse the experimental $^{13}$C NMR spectra acquired for different $^1$H polarizations and herein present a straightforward approach to indirectly quantify the $^1$H polarization based on the $^{13}$C NMR peak normalized deviation of the centre of gravity (CoG). $^1$H polarization was observed to increase with an increasing $^{13}$C NMR peak CoG deviation (case of positive microwave irradiation).

## 2. Methods

### 2.1. Sample Preparation

A solution of 3 M [2-$^{13}$C]sodium acetate in the glass-forming mixture $H_2O/D_2O$/glycerol-$d_8$ (1/3/6 $v/v/v$) was doped with 50 mM TEMPOL radical (all compounds purchased from *Sigma Aldrich*) and sonicated for ~10 minutes. Paramagnetic TEMPOL radicals were chosen to polarize $^1$H spins most efficiently under our *d*DNP conditions.

### 2.2. Sample Freezing

A 100 $\mu$L volume of the above sample was pipetted into a Kel-F sample cup and inserted into a 7.05 T prototype *Bruker Biospin* polarizer equipped with a specialized *d*DNP probe, including a background-free radiofrequency (*rf*) coil insert (Elliott et al, 2021), running *TopSpin 3.5* software. The sample temperature was reduced to 1.2 K by submerging the sample in liquid helium and reducing the pressure of the variable temperature insert (VTI) towards ~0.7 mbar.

### 2.3. Dynamic Nuclear Polarization

The 100 $\mu$L of sample was polarized by applying microwave irradiation at $f_{\mu w}$ = 197.616 GHz (positive lobe of the DNP enhancement profile) or $f_{\mu w}$ = 198.192 GHz (negative lobe of the DNP enhancement profile) with triangular frequency modulation (Bornet et al, 2014) of amplitude $\Delta f_{\mu w}$ = $\pm$ 136 MHz or $\Delta f_{\mu w}$ = $\pm$ 112 MHz, respectively, and rate $f_{mod}$ = 0.5 kHz at a power of ca. 125 mW at the output of the microwave source (value given by the provider of our microwave source *VDI/AMC 705*) and ca. 30 mW reaching the DNP cavity (evaluated by monitoring the helium bath pressure, see Section 2.4), which were optimized prior to commencing experiments to achieve the highest possible level of $^1$H polarization.

### 2.4. Microwave Power Evaluation

The microwave power reaching the DNP cavity was determined by comparison with the heating from a resistor in the liquid helium bath and calibrating how much the bath pressure increases vs. microwave power. In practice, the measurement was performed as follows:

(*i*) The VTI was filled with liquid helium and pumped down to 0.65 mbar, corresponding to 1.2 K;

(*ii*) The change of pressure when turning on a resistive heater or the microwave source for 120 s was monitored. The pressure plateaus after approximatively 60 s;

(*iii*) The pressure difference between the base pressure and that under the effect of the resistive heater or the microwave source

$\Delta P_{mbar}$ is calculated.

All measurements were performed ensuring that the liquid helium level in the VTI was not varying by more than a few

centimetres: the microwave cavity was immersed under 5-10 cm of liquid helium. The measurements performed using the resistive

heater with power $P_{heater}$ are used to plot a calibration curve $P_{heater}$ vs. $\Delta P_{mbar}$ with slope $a$. The deposited microwave power in the

cavity is then obtained by computing $P_{microwave} = a\Delta P_{mbar}$.

**2.5. Polarization Build-Ups**

To monitor $^{13}$C NMR spectral lineshapes with satisfactory signal-to-noise ratios (SNRs), $^{13}$C polarization must first be built-up by

using a succession of optimized cross-polarization (CP) contact *rf*-pulses. Then, to observe changes in the lineshape of $^{13}$C NMR

spectra acquired as the $^1$H polarization builds up from the thermal to DNP equilibrium, we employed a series of $^1$H saturating *rf*-

pulses followed by microwave activation, a small flip-angle *rf*-pulse and $^{13}$C NMR signal detection, as shown by the *rf*-pulse

sequence in Figure 1. The build-up of $^{13}$C polarization throughout the microwave irradiation period was tracked by engaging the

following experimental procedure:

(*i*) A saturating sequence of 90° *rf*-pulses with alternating phases separated by a short delay (typ. 11 ms) repeated *n* times (typ.

*n* = 50) kills residual magnetization on both *rf*-channels;

(*ii*) The microwave source becomes active and $^1$H polarization builds up;

(*iii*) The $^{13}$C Zeeman magnetization trajectory is minimally perturbed by the application of a small flip-angle *rf*-pulse (typ. $\beta$ =

3.5°) used for detection, which is then followed by a short acquisition period (typ. $t_{FID}$ = 1 ms);

(*iv*) $^1$H DNP builds up during a time $t_{DNP}^1$ (typ. $t_{DNP}^1$ = 30 s);

(*v*) Stages *iii-iv* are cycled *m* times (typ. *m* = 6) in order to monitor the evolution of the $^{13}$C polarization (between CP steps);

(*vi*) The microwave source is gated and a delay of duration $t_G$ = 0.5 s occurs, see Section 2.6, thus permitting the electron spins

to relax to their highly polarized thermal equilibrium state before the next CP step (Bornet et al, 2016);

(*vii*) Two synchronized adiabatic half-passages (AHPs) simultaneously produce transverse magnetization for all pulsed spin

species;

(*viii*) The nuclear magnetization is subsequently spin-locked on both *rf*-channels (typically by a high power *rf*-pulse with a

nutation frequency on the order of 15 kHz and a duration between 1-10 ms) and $^1$H→$^{13}$C polarization transfer occurs (Bornet et al,

2016);

(*ix*) A second pair of harmonized AHPs (operating with reverse chronology) restores Zeeman magnetization on each *rf*-channel;

(*x*) Stages *ii-ix* are repeated in *L* units (typ. *L* = 8) to periodically transfer $^1$H Zeeman polarization to $^{13}$C nuclear spins;

(*xi*) A second saturating sequence of 90° *rf*-pulses with alternating phases separated by a short delay (typ. 11 ms) repeated *n*

120    times (typ. *n* = 50) kills residual magnetization on the $^1$H *rf*-channel only;

(*xii*) The microwave source reactivates;

(*xiii*) The $^{13}$C Zeeman magnetization trajectory is minimally perturbed by the application of a small flip-angle *rf*-pulse (typ. $\beta$

= 3.5°) used for detection, which is then followed by a short acquisition period (typ. $t_{FID}$ = 1 ms);

(*xiv*) $^1$H DNP builds up during a time $t_{DNP}^2$ (typ. $t_{DNP}^2$ = 5 s);

(*xv*) Stages *xiii-xiv* are cycled *p* times (typ. *p* = 80) to monitor the evolution of the $^{13}$C NMR spectra as a function of the $^1$H

polarization build-up with sufficient SNR.

Further details regarding multiple-contact CP *rf*-pulse sequence operation are given elsewhere (Elliott et al, 2021b). It should

be stressed that the use of CP is purely optional, and in most cases its use will be dictated by the *rf*-hardware available. We use CP

.29 here simply as a means to offer greater SNRs for $^{13}$C NMR signal detection. Given the level of sample deuteration, at 6.7 T and

.30 with microwave modulation suitable SNRs can also be achieved with direct $^{13}$C DNP (Chen et al., 2013).

.31    Since it is unlikely that the $^{13}$C NMR lineshape is significantly influenced by the $^{13}$C polarization, we can afford not to diminish

.32 the $^{13}$C NMR signal intensity by a sequence of $^{13}$C saturating *rf*-pulses on the $^{13}$C *rf*-channel at stage *xi* to maintain high SNRs. The

.33 small *rf*-pulse flip angles are necessary to preserve the $^{1}$H and $^{13}$C polarizations throughout the course of the build-up experiment.

.34

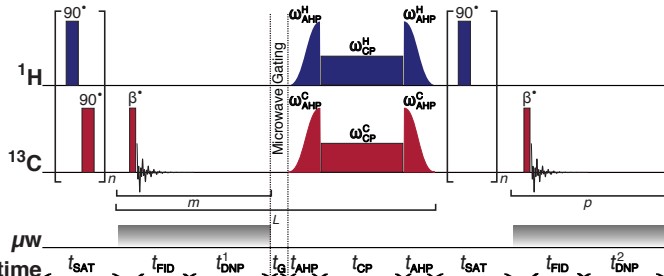

.35

.36

.37 **Figure 1: Schematic representation of the *rf*-pulse sequence used to accrue $^{13}$C polarizations and monitor $^{13}$C lineshapes as a function of the $^{1}$H polarization.**

.38 **The experiments used the following key parameters chosen to maximize the efficiency of the *rf*-pulse sequence: $n = 50$; $\beta = 3.5°$; $m = 6$; $t_{DNP}^{1} = 30$ s; $L = 8$;**

.39 **$t_G = 0.5$ s; $p = 80$; and $t_{DNP}^{2} = 5$ s. AHP = adiabatic half-passage. AHP sweep width = 100 kHz. The $\pi/2$ saturating *rf*-pulses used an empirically optimized**

.40 **thirteen-step phase cycle to remove residual magnetization at the beginning of each experiment: {0, $\pi/18$, $5\pi/18$, $\pi/2$, $4\pi/9$, $5\pi/18$, $8\pi/9$, $\pi$, $10\pi/9$, $13\pi/9$,**

.41 **$\pi/18$, $5\pi/3$, $35\pi/18$}. The resonance offset was placed at the most intense peak of the $^{1}$H and $^{13}$C NMR spectra.**

.42

.43 **2.6. Microwave Gating**

.44

.45 Microwave gating was employed shortly before and during CP experiments to allow the electron spin ensemble to return to a highly

.46 polarized state, which happens on the timescale of the longitudinal electron relaxation time (typ. $T_{1e} = 100$ ms with $P_e = 99.93\%$

.47 under our experimental *d*DNP conditions) (Bornet et al, 2016). Microwave gating hence provides a way to strongly attenuate

.48 paramagnetic relaxation, and consequently the $^{1}$H and $^{13}$C $T_{1\rho}$ relaxation time constants in the presence of an *rf*-field are extended

.49 by orders of magnitude. This allows spin-locking *rf*-pulses to be much longer, which significantly increases the efficiency of nuclear

.50 polarization transfer.

.51

.52 **3. Results**

.53

.54 **3.1. $^{13}$C CP Build-Ups and Decays**

.55

.56 The CP build-up curves for the $^{13}$C polarizations $P_C$ as a function of the $^{1}$H DNP time $t_{DNP}$ for both positive and negative microwave

.57 irradiation are shown in Figure 2. The $^{13}$C polarizations $P_C$ were accrued by employing the *rf*-pulse sequence shown in Figure 1.

.58 The $^{13}$C polarizations $P_C$ ultimately reached $P_C \simeq 40.6\%$ and $P_C \simeq -46.8\%$ after 8 CP transfers and 24 minutes of positive and

.59 negative microwave irradiation, respectively. The achieved levels of $^{13}$C polarization $P_C$ are lower than those previously reported

.60 in the literature (Bornet et al, 2016), but were not further optimized since only the $^{13}$C NMR lineshape was of interest in this study

.61 as a probe for absolute $^{1}$H polarization. This is inconsequential for the current study since sufficient SNRs on the order of ~965 and

.62 ~1244 were achieved for the cases of positive and negative microwave irradiation, respectively. After this point, *i.e.,* beyond the

.63 vertical dashed line ($^{1}$H DNP time = 24 mins), a slow and partial decay in the $^{13}$C NMR signal intensity towards a pseudo-

.64 equilibrium is observed, see Figure 2. This $^{13}$C NMR signal decay is not a problem in general since the $^{13}$C NMR signal remains

.65 sufficiently intense as to allow clear measurement of the $^{13}$C NMR lineshape with high accuracy.

.66

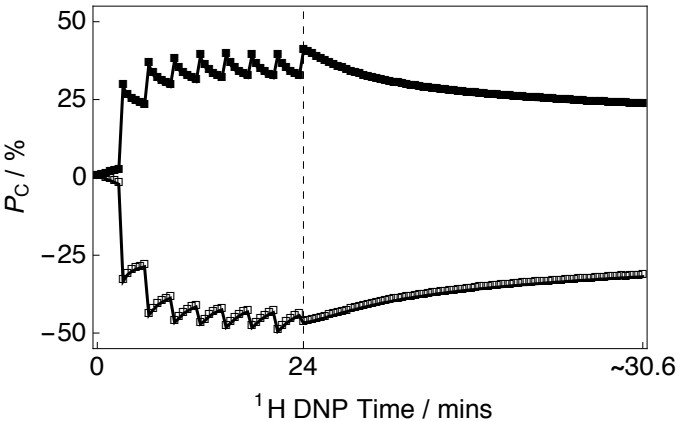

**Figure 2: Experimental $^{13}$C polarization $P_C$ CP build-up curves and subsequent $^{13}$C signal decays as a function of $^1$H DNP time acquired at 7.05 T ($^1$H nuclear Larmor frequency = 300.13 MHz, $^{13}$C nuclear Larmor frequency = 75.47 MHz) and 1.2 K with a single transient per data point. The presented data were acquired by using the *rf*-pulse sequence depicted in Figure 1. Black filled squares: Positive microwave irradiation; Black empty squares: Negative microwave irradiation. The vertical dashed line denotes the $^1$H DNP time at which the $^1$H NMR signal was destroyed by a second series of saturating *rf*-pulses (as shown by the *rf*-pulse sequence illustrated in Figure 1).**

## 3.2. $^{13}$C NMR Spectra

Figure 3 shows the relevant part of the experimental $^{13}$C NMR spectra acquired with a small flip angle *rf*-pulse ($\beta = 3.5°$) at two different $^1$H DNP times. The $^{13}$C NMR spectra in Figure 3 were acquired by using the *rf*-pulse sequence shown in Figure 1. The initial $^{13}$C NMR spectrum (acquired at 24 mins) is a single peak with a linewidth at full-width half-maximum height (FWHM) of ~10.9 kHz. The $^{13}$C NMR lineshape is relatively symmetrical and has no obvious defining features, see Figure 3a. Small peak contributions to the $^{13}$C NMR spectrum are observed towards the baseline, including one environment shifted as much as ca. -300ppm. This spectrum corresponds to a low level of $^1$H polarization ($|P_H| \simeq 0\%$).

However, the $^{13}$C NMR spectra become more complicated and gain sharper spectral features at extended $^1$H DNP times, see Figures 3b and 3c. At ~30.6 mins, the $^{13}$C NMR spectra are comprised of (at least) two main resonances with differing NMR signal intensities. In the case of positive microwave irradiation (Figure 3b), the frequency separation between the two most intense $^{13}$C NMR peaks is ~8.4 kHz and the linewidth at FWHM is ~17.7 kHz. It is interesting to note that the $^{13}$C NMR spectra acquired in the cases of positive (Figure 3b) and negative (Figure 3c) microwave irradiation do not have the same overall profile at long $^1$H DNP times. These spectra correspond to much higher levels of $^1$H polarization ($|P_H| \gtrsim 55\%$).

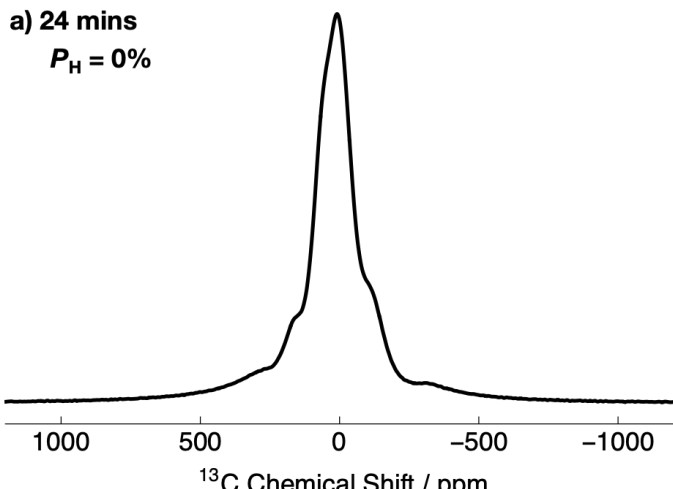

**a) 24 mins**
   $P_H = 0\%$

1000   500   0   −500   −1000

$^{13}$C Chemical Shift / ppm

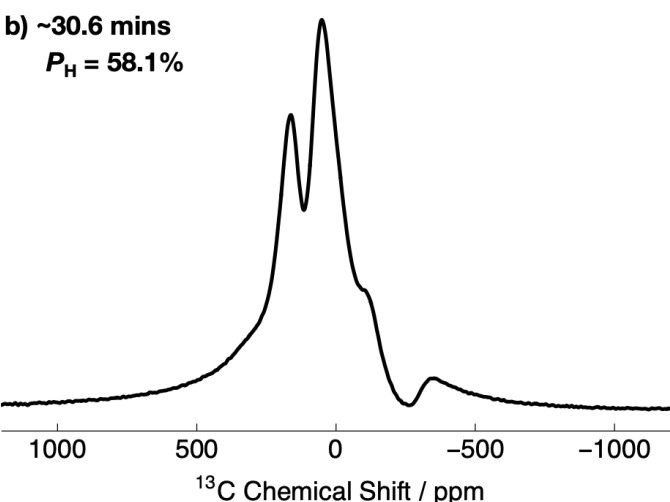

**b) ~30.6 mins**
   $P_H = 58.1\%$

1000   500   0   −500   −1000

$^{13}$C Chemical Shift / ppm

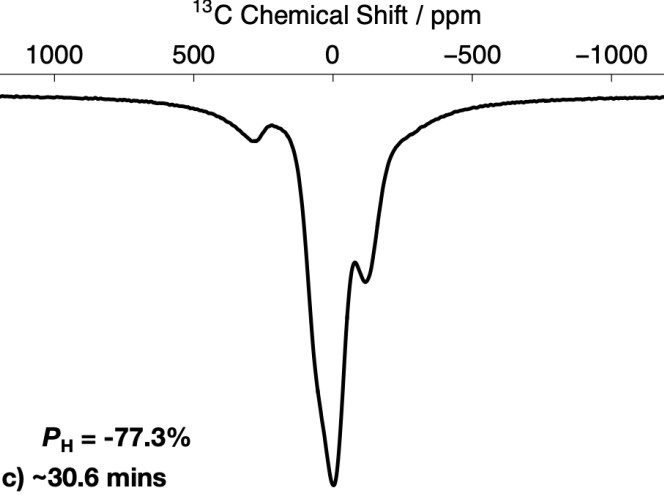

$^{13}$C Chemical Shift / ppm

1000   500   0   −500   −1000

$P_H = -77.3\%$
**c) ~30.6 mins**

**Figure 3: Relevant portions of the experimental $^{13}$C NMR spectra belonging to the $^{13}$C-labelled methyl group ($^{13}$CH$_3$) of [2-$^{13}$C]sodium acetate acquired at 7.05 T ($^1$H nuclear Larmor frequency = 300.13 MHz, $^{13}$C nuclear Larmor frequency = 75.47 MHz) and 1.2 K with a single transient (*rf*-pulse flip angle = 3.5°) at two different $^1$H DNP times. The labels indicate the $^1$H DNP times at which the spectra were recorded. The timings coincide with those shown in Figure 2. The $^{13}$C NMR spectra were acquired by using the *rf*-pulse sequence depicted in Figure 1. (a) No microwave irradiation; (b) Positive microwave irradiation; and (c) Negative microwave irradiation. All $^{13}$C NMR spectra have been scaled to yield the same maximum intensity.**

### 3.3. $^{13}$C NMR Peak Normalized Centre of Gravity Deviation vs. $^1$H Polarization

The DNP build-up curve for the $^1$H polarization $P_H$ as a function of the $^1$H DNP time for positive microwave irradiation is shown in Figure 4. More details regarding how to acquire such build-up curves are given in the following reference (Bornet et al, 2016). The $^1$H polarization build-up curve was found to have a stretched exponential behaviour, and the experimental data are well fitted with a stretched exponential function using a $^1$H DNP build-up time constant denoted $\tau_{DNP}^+$. Stretched exponential function: $A(1-\exp\{-(t/\tau_{DNP}^+)^\beta\})$, where A is a constant, $\tau_{DNP}^+$ is the $^1$H DNP build-up time constant extracted from the above fitting procedure and $\beta$ is the breadth of the distribution of $^1$H DNP build-up time constants. The mean $^1$H DNP build-up time constant $\langle\tau_{DNP}^+\rangle$ is calculated as follows: $\langle\tau_{DNP}^+\rangle = \tau_{DNP}^+\Gamma(1/\beta)/\beta$, where $\Gamma(1/\beta)$ is the gamma function. A similar $^1$H polarization build-up curve for the case of negative microwave irradiation, with parameters $\tau_{DNP}^-$ and $\langle\tau_{DNP}^-\rangle$, is shown in the Supplement.

The sample polarized to $P_H \simeq$ -77.3% ($^1$H DNP time $\simeq$ 30.6 mins) by employing negative microwave irradiation with a $^1$H DNP build-up time constant of $\langle\tau_{DNP}^-\rangle = 122.0 \pm 0.4$ s ($\beta = 0.87$). A reduced $^1$H polarization of $P_H \simeq$ 58.1% was reached (at $^1$H DNP time $\simeq$ 30.6 mins) by using positive microwave irradiation. The $^1$H DNP build-up time constant was much shorter in this case: $\langle\tau_{DNP}^+\rangle = 80.2 \pm 0.3$ s ($\beta = 0.77$).

The $^{13}$C NMR lineshapes presented in Figure 3 are complicated and so it is desirable to construct a parameter which can describe the $^1$H polarization $P_H$, be robust with respect to field inhomogeneities and easily applied to any lineshape. Figure 4 therefore also displays the $^{13}$C NMR peak CoG deviation $\delta_{\omega_0}$ for sample **I** as a function of the $^1$H DNP time for the case of positive microwave irradiation. The $^{13}$C NMR peak CoG normalized deviation $\delta_{\omega_0}$ is defined as:

$$\delta_{\omega_0} = \frac{M_{asym}}{LW_0} \quad (1)$$

where $M_{asym}$ is denoted as the first moment of asymmetry and corresponds to the following quantity:

$$M_{asym} = \int_{-\infty}^{\infty}(\omega - \omega_0(P_H = 0\%))\,f(\omega)\,d\omega \quad (2)$$

The first moment of asymmetry $M_{asym}$ is based on a calculation whereby the CoG of the $^{13}$C NMR peak $\omega_0$ is held constant at $\omega_0(P_H = 0\%)$, i.e., the $^{13}$C NMR peak CoG corresponding to when the $^1$H polarization $P_H$ is zero. The CoG of the $^{13}$C NMR peak $\omega_0$ is calculated as:

$$\omega_0 = \int_{-\infty}^{\infty}\omega\,f(\omega)\,d\omega \quad (3)$$

where the intensities of the $^{13}$C NMR peaks are normalized:

$$\int_{-\infty}^{\infty} f(\omega)\,d\omega = 1 \quad (4)$$

where $\omega$ is the resonance frequency and $f(\omega)$ is the peak intensity at $\omega$. The procedure outlined above ensures that $M_{asym} = 0$ at $P_H = 0\%$ such that the described approach can be readily generalized to any lineshape. The quantity $LW_0$ is a measure of the linewidth of the $^{13}$C NMR peak in the case of $P_H = 0\%$:

$$LW_0 = \sqrt{\int_{-\infty}^{\infty}(\omega(P_H = 0\%) - \omega_0(P_H = 0\%))^2\,f(\omega(P_H = 0\%))\,d\omega} \quad (5)$$

*i.e.*, the square root of the second moment at $P_H = 0\%$. This factor establishes a $^{13}$C NMR peak CoG deviation $\delta_{\omega_0}$ (defined in

Equation 1) which is a normalized and dimensionless quantity.

Figure 4 indicates that at longer $^1$H DNP times, where the $^1$H polarization $P_H$ is higher, there is a greater $^{13}$C NMR peak CoG

normalized deviation $\delta_{\omega_0}$. Similar curves to those presented in Figure 4 for the case of negative microwave irradiation are shown

in the Supplement. It should be noted that the curve profiles and final values of $\delta_{\omega_0}$ are not mirror images of each other. This is

also reflected in the $^{13}$C NMR spectra acquired at ~30.6 mins, see Figure 3. The rate of change in the value of $\delta_{\omega_0}$ during the first

~100 s of Figure 4 indicates a more rapid change in the $^1$H polarization $P_H$. This coincides with the starkest changes in $^{13}$C NMR

lineshape, see the Supplement.

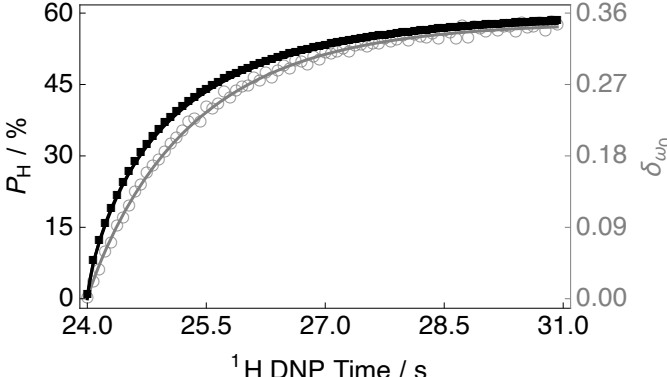

**Figure 4: Experimental $^1$H polarization $P_H$ DNP build-up curve (black filled squares and left-hand axis) and $^{13}$C NMR peak CoG normalized deviation**

**$\delta_{\omega_0}$ (grey empty circles and right-hand axis) as a function of the $^1$H DNP time acquired at 7.05 T ($^1$H nuclear Larmor frequency = 300.13 MHz, $^{13}$C nuclear**

**Larmor frequency = 75.47 MHz) and 1.2 K with a single transient per data point for the case of positive microwave irradiation. The timings coincide with**

**those shown in Figure 2. The black solid line indicates the best fit of the experimental data points for the $^1$H polarization $P_H$ DNP build-up curve, and has**

**the corresponding fitting function: A(1-exp{-(t/$\tau_{DNP}^{\pm}$)$^{\beta}$}). Mean $^1$H DNP build-up time constant: $\langle\tau_{DNP}^{+}\rangle$ = 80.2 ± 0.3 s.**

The $^{13}$C NMR peak CoG normalized deviation $\delta_{\omega_0}$ as a function of the $^1$H polarization $P_H$ for positive microwave irradiation is

shown in Figure 5. The $^1$H polarization $P_H$ increases with an increasing $^{13}$C NMR peak CoG normalized deviation. The experimental

data were fitted with a phenomenological relationship of the kind: $P_H(\delta_{\omega_0}) = A \times \delta_{\omega_0}^{\beta}$, where $P_H(\delta_{\omega_0})$ is the $^1$H polarization as a

function of the $^{13}$C NMR peak CoG normalized deviation $\delta_{\omega_0}$, $\beta$ is the order of the polynomial fit and $A$ is a scaling factor. The

phenomenological function is simply used to correlate the $^{13}$C NMR peak CoG normalized deviation $\delta_{\omega_0}$ with the $^1$H polarization

$P_H$. The best fit values of the phenomenological function to the experimental data over the range of $^{13}$C NMR peak CoG normalized

deviations shown in Figure 5 are given in the caption.

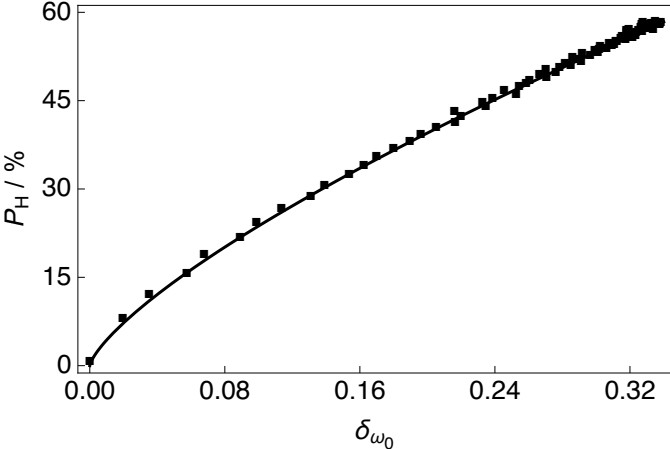

Figure 5: Experimental $^1$H polarizations $P_H$ as a function of the $^{13}$C NMR peak CoG normalized deviation $\delta_{\omega_0}$ acquired at 7.05 T ($^1$H nuclear Larmor frequency = 300.13 MHz, $^{13}$C nuclear Larmor frequency = 75.47 MHz) and 1.2 K with a single transient per data point for the case of positive microwave irradiation. The experimental data were fitted with a phenomenological function: $P_H(\delta_{\omega_0}) = A \times \delta_{\omega_0}^{\beta}$. Best fit values: $A = 129.1\% \pm 0.8\%$; $\beta = 0.736 \pm 0.005$. The absolute $^1$H polarizations $P_H$ were measured by comparison with a thermal equilibrium $^1$H NMR signal.

## 4. Discussion

As discussed in Section 3.3 above, the CoG normalized deviation $\delta_{\omega_0}$ of the peaks in the $^{13}$C NMR spectrum indirectly provide the level of $^1$H polarization $P_H$, see Figure 5. It is unlikely that a uniform spin temperature between the $^1$H and $^{13}$C nuclear spin reservoirs is reached at any time during the experiment presented in Figure 1, but as long as a uniform spin temperature is achieved within the $^1$H nuclear spin reservoir then the methodology presented above holds. It should be noted that the order of the polynomial fit $\beta$ shown in Figure 5 is likely to be influenced by the capabilities of the *rf*-probe, such as the *rf*-pulse homogeneity, and it is therefore recommended that (if possible) users implement similar measurements on their own experimental setups, rather than simply reusing the value presented here. In this way, any laboratory can adopt the procedure and reproduce the result.

Once the $^{13}$C NMR peak CoG normalized deviation $\delta_{\omega_0}$ falls below zero the $^1$H polarization $P_H$, rapidly drops towards negative $^{13}$C NMR peak CoG normalized deviations $\delta_{\omega_0}$ (with decreasing $^1$H polarizations $P_H$). This result implies that the NMR peak CoG normalized deviation $\delta_{\omega_0}$ is less sensitive to negative microwave irradiation. This change in sensitivity of the $^{13}$C NMR peak CoG normalized deviations $\delta_{\omega_0}$ to positive and negative microwave irradiation is also evident in the $^{13}$C NMR spectra, see Figure 3 and the Supplement. This is likely associated with: (*i*) $^{13}$C NMR spectra at negative levels of $^1$H polarization have lineshapes with less pronounced features, *i.e.,* partially unresolved peaks; and (*ii*) the $^{13}$C NMR lineshape changes less dramatically as a function of negative $^1$H polarization. These points could both be related to NMR line narrowing due to radiation damping in the case of negative microwave irradiation (Mao and Ye, 1997; Krishnan and Murali, 2013).

$^1$H polarizations in the range of $P_H \lesssim 30\%$ correspond to those typically accrued by $^1$H DNP build-up experiments at liquid helium temperatures of 3.8-4.2 K ($P_H$ = 0-30%). These results indicate that the $^{13}$C NMR peak CoG normalized deviation $\delta_{\omega_0}$ can therefore also be used to infer $^1$H polarizations $P_H$ accurately at elevated temperatures. However, the presence of methyl group rotation at temperatures above 1.2 K is likely to somewhat average the $^1$H-$^{13}$C dipolar couplings and could lead to a different trend compared with the fit presented in Figure 5 (Latanowicz, 2005).

One possible contribution to the inflexion in the fit of the $^{13}$C NMR peak CoG normalized deviations $\delta_{\omega_0}$ at low levels of $^1$H polarization $P_H$ is the presence of strong polarization gradients or highly polarized clusters of nuclear spins located within specific radii of the electron spins within the sample at short $^1$H DNP times, which would lead to a non-uniform spin temperature. This contribution is expected to be minor.

The decay of $^{13}$C polarization during the $^1$H DNP build-up interval $t_{DNP}^2$ shown in Figure 2 occurs when the microwave source is active and the $^{13}$C nuclear spin ensemble relaxes towards the spin temperature it would have achieved in the case of direct $^{13}$C DNP, *i.e.*, no CP. This $^{13}$C polarization decay is a combination of three factors: (*i*) the microwaves are active and hence polarization is diminishing towards the low DNP equilibrium of the $^{13}$C nuclear spins with TEMPOL as the polarizing agent; (*ii*) the $^{13}$C nuclear spins are being actively pulsed, although minimally, every 5 s, which leads to an accumulative loss of $^{13}$C NMR signal intensity over many minutes; and (*iii*) the radical concentration and temperature are in an optimal range for thermal mixing (Guarin et al, 2017) and since the $^{13}$C spins are polarized whilst the $^1$H spins are saturated the two nuclear pools most likely exchange energy via the electron non-Zeeman reservoir, which influences the time evolution of the $^{13}$C magnetization until the $^1$H spins achieve the same spin temperature. The difference in the $^{13}$C polarizations $P_C$ at $^1$H DNP time = 24 mins for positive and negative microwave irradiation is associated with the $^1$H polarization build-ups and the performance efficiency of the multiple-contact CP *rf*-pulses, see the Supplement.

The $^{13}$C NMR lineshapes of [2-$^{13}$C]sodium acetate shown in Figure 3 have features which mainly originate from $^{13}$C chemical shift anisotropy (CSA) (max. ~1.5 kHz at our magnetic field of 7.05 T) and $^1$H-$^{13}$C dipolar couplings (typ. -22.7 kHz) that are affected by possible methyl group rotation. Since the $^{13}$C CSA is negligible with respect to the $^1$H-$^{13}$C dipolar couplings, it is assumed that the $^1$H-$^{13}$C dipolar couplings play the key role in the $^{13}$C NMR lineshape of [2-$^{13}$C]sodium acetate. The smaller $^{13}$C NMR peak contributions observed near the baseline in Figure 3a likely correspond to different chemical environments within the sample which are being polarized on different time scales.

The values of $\delta_{\omega_0}$, $P_H$ and the order of the polynomial fit $\beta$ presented in Figure 5 are likely to depend to a small degree on the solvent constituents. In the case of our sample, the glycerol-$d_8$ present in the *d*DNP glassing matrix yields an approximate $^{13}$C concentration of ~410 mM at natural abundance, which is ~14% of the total $^{13}$C NMR signal. Under microwave irradiation, the natural abundance $^{13}$C spins of glycerol-$d_8$ will be polarized with their own build-up rate and maximum polarization, and although deuterated glycerol-$d_8$ can also be polarized by $^1$H-$^{13}$C CP (Vuichoud et al, 2019). As such, these contributions could impact the $^{13}$C NMR peak intensities, which would go some way to explaining why the $^{13}$C NMR spectra are not of the same overall profile under positive and negative microwave irradiation at long proton DNP times, see Figures 3b and 3c. It is also possible that the dipolar couplings and CSA interactions manifest differently under positive and negative microwave irradiation, and that there is a preferred energy state for coupling to positive and negative $^1$H polarizations $P_H$ leading to non-identical $^{13}$C NMR spectra.

The NMR spectra presented in Figure 3 were acquired for the cases of high $^{13}$C SNRs, the largest of which is ca. 1244. In the event that CP cannot be efficiently implemented, and the acquired $^{13}$C NMR signal is weak, we anticipate that the method is robust with respect to a few kilohertz of Lorentzian line broadening, which can be used to improve the experimental SNR. The value of the $^{13}$C NMR peak CoG normalized deviation $\delta_{\omega_0}$ is, however, likely to be sensitive to changes in phase, and this should therefore be taken into account before comparing experimental results to any calibration curves similar to those presented in Figure 5. It is also possible that additional phase corrections may help the trend shown in Figure 5 move closer to a linear fit for values of $\delta_{\omega_0} <$ 0.02.

The results of this study suggest that other $^{13}$C-labelled molecules which might display distinct solid-state $^{13}$C NMR spectra, such as [1-$^{13}$C]sodium formate and other $^{13}$CH$_3$, or $^{13}$CH$_2$, group bearing molecular candidates (presence of a strong $^1$H-$^{13}$C dipolar coupling), could also be used as indirect $^1$H polarization meters. To effectively polarize both $^1$H and $^{13}$C nuclear spins, future experiments could use a tailored mixture of radical species, in certain cases. Clearly, at low levels of $^1$H polarization $P_H$ the lower intensity resonance is unresolved and polluted by the more intense peak, and as such; the presented analysis could be further improved by considering Voigt fits of the complicated $^{13}$C NMR spectra, but since there are a number of resonances to consider this route would lead us away from our simple pedagogical approach.

**5 Conclusions**

We have demonstrated that $^{13}$C NMR lineshape polarimetry of [2-$^{13}$C]sodium acetate can be implemented to indirectly infer the $^1$H polarization of the $^{13}$CH$_3$ group nuclear spins and potentially the whole sample if the constituents of which are sufficiently homogeneously mixed. An easy to implement protocol based on the normalized deviation of the centre of gravity of the $^{13}$C peaks was employed and a simple relationship with $^1$H polarization was found. This approach is complementary to traditional methods of measuring $^1$H polarization, in suitable circumstances, and could be useful in situations where measurements of $^1$H polarizations prove difficult, *e.g.,* due to radiation damping (Mao and Ye, 1997; Krishnan and Murali, 2013), which can also likely impact the experimental data and order of the polynomial fit shown in Figure 5. Other appropriate cases for potential implementation include: (*i*) the lack of a $^1$H *rf*-coil; (*ii*) the presence of large background signals; and (*iii*) the absence of a thermal equilibrium spectrum. The approach presented here works well for traditional *d*DNP-compatible sample formulations but future studies employing fully deuterated *d*DNP solutions could provide $^{13}$C NMR lineshapes with more distinct features.

## Acknowledgements

The authors gratefully acknowledge *Bruker Biospin* for providing the prototype *d*DNP polarizer, and particularly Dmitry Eshchenko, Roberto Melzi, Marc Rossire, Marco Sacher and James Kempf for scientific and technical support. The authors additionally acknowledge Catherine Jose and Christophe Pages for use of the ISA Prototype Service, Stéphane Martinez of the UCBL mechanical workshop for machining parts of the experimental apparatus, Aurélien Bornet for preliminary works, Christian Bengs for useful discussions and the reviewers for their comments which led to the improvement of this article.

## Financial Support

This research was supported by ENS-Lyon, the French CNRS, Lyon 1 University, the European Research Council under the European Union's Horizon 2020 research and innovation program (ERC Grant Agreements No. 714519 / HP4all and Marie Skłodowska-Curie Grant Agreement No. 766402 / ZULF).

## Author Contributions

SJE conceived the idea, performed experiments, processed the data and wrote the manuscript, QC assisted with experiments and data processing, and provided useful advice, and SJ provided informative guidance, supportive feedback and contributed to the manuscript.

## Data Availability

Experimental data are available upon request from the corresponding author.

## Competing Interests

The authors declare no competing interests.

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
