# Peer review of "Solid-State 1H Spin Polarimetry by 13CH3 Nuclear Magnetic Resonance"

_Magnetic Resonance, 2021_

## Community Comment (CC1)

The paper on "Solid-State 1H Spin Polarimetry by 13CH3 Nuclear Magnetic Resonance"
by Stuart J. Elliott, Quentin Stern (or should it be Quentin Chappuis, as in the
Supplement?) and Sami Jannin raises all sorts of interesting questions.

When the authors write "The frequency of the dashed line corresponds to the minimum
between the two peaks at high levels of 1H polarization," they discuss the two peaks
phenomenologically as if they constituted a doublet. Are they implicitly referring to a doublet
due to the heteronuclear dipole-dipole coupling in the Hamiltonian?

Is this splitting only seen in methyl groups? Do these methyl groups have to rotate freely all
the way down to sample temperatures near 1.2 K? Is that the case for methyl groups in
acetate? Do you see similar effects in molecules such as gamma picoline that are known to
have very low rotation barriers? If the methyl rotation in acetate is frozen, why are
the methyl signals in acetate any different from those of other rigid chemical moieties like
CH2 groups? You suggest that similar effects can be observed in molecules "such as [1-
13C]sodium formate", although there are no methyl groups in formate.

Can you explain your observation "where the [proton] polarization is higher, the peak
intensities become more equal"? Can you explain why "[proton] polarizations (…) were
observed to *decrease* linearly with *increasing* 13C peak asymmetries"? As you wrote,
this behaviour is opposite to what was observed by Mammoli et al. and by Aghelnejad et al.
It seems that polarisation and asymmetry should increase together.

What do you expect to see in the presence of "solid-state methyl group AE population
imbalances at low temperatures"?

For practitioners of DNP, it would be nice to know more details about how you determined
the "power of ca. 80 $P\mu w$ = 125 mW at the output of the microwave source and ca. $P\mu w$ = 30
mW reaching the DNP cavity (evaluated by monitoring the helium bath pressure.)" Is the first
number given by the manufacturer of the microwave source? How did you convert pressure
changes into mW? How do you calibrate this empirical relationship?

The statement "the 1H and 13C [transverse] relaxation time constants in the presence of an
rf-field are extended by orders of magnitude" seems a bit exaggerated. Bornet et al. have
shown experimentally that T_1rho can be extended by a factor 5 or so when the microwaves
are switched on.

I hope that these naive comments will contribute to make the "discussions" of "Magnetic
Resonance" more lively, and may help to reduce inhibitions and "open the gates" to further
comments from interested readers.

---

## Author Comment (AC1)

Comment 1: Geoffrey Bodenhausen.

*The paper on "Solid-State 1H Spin Polarimetry by 13CH3 Nuclear Magnetic Resonance" by Stuart J. Elliott, Quentin Stern (or should it be Quentin Chappuis, as in the Supplement?) and Sami Jannin raises all sorts of interesting questions.*

The surname of our colleague has been amended in the supplement. Thank you for pointing this out to us.

*When the authors write "The frequency of the dashed line corresponds to the minimum between the two peaks at high levels of 1H polarization," they discuss the two peaks phenomenologically as if they constituted a doublet. Are they implicitly referring to a doublet due to the heteronuclear dipole-dipole coupling in the Hamiltonian?*

We believe that the strong HC dipolar coupling is involved in this phenomenon. We also agree that more information should be given about the origin of this asymmetry, and we will therefore add a paragraph listing the different interactions considered:

"The 13C NMR lineshape of [2-13C]sodium acetate has features which mainly originate from 13C chemical shift anisotropy (CSA) (typ. ~1.5 kHz at our magnetic field of 7.05 T) and 1H-13C dipolar couplings (typ. -22.7 kHz) that are affected by possible methyl group rotation. Since the 13C CSA is negligible with respect to the 1H-13C dipolar couplings, it is assumed that the 1H-13C dipolar couplings play the key role in the 13C NMR lineshape of [2-13C]sodium acetate."

*Is this splitting only seen in methyl groups? Do these methyl groups have to rotate freely all the way down to sample temperatures near 1.2 K? Is that the case for methyl groups in acetate? Do you see similar effects in molecules such as gamma picoline that are known to have very low rotation barriers? If the methyl rotation in acetate is frozen, why are the methyl signals in acetate any different from those of other rigid chemical moieties like CH2 groups? You suggest that similar effects can be observed in molecules "such as [1- 13C]sodium formate", although there are no methyl groups in formate.*

We have only seen such interesting lineshapes, *i.e.*, ones with similarly pronounced features, for [2-13C]acetate thus far. Other 13C-labelled methyl groups, such as gamma-picoline, and 13C-labelled methylene groups might also exhibit exotic spectra under our experimental *d*DNP conditions, and it is also possible that such effects could be observed in [1-13C]formate, if indeed one of the main requirements is the presence of a strong HC dipolar coupling. However, we have not yet investigated other suitable molecular candidates. To our knowledge, the observed asymmetry does not require methyl group rotation, and only a strong HC dipolar coupling. Particularly fast methyl group rotation is likely to average HC dipolar couplings and is likely to work against the phenomena observed here. As can be seen from J.-N. Dumez et al., *J. Phys. Chem. Lett.* **2017**, 8, 3549-3555, quantum rotor induced polarization (QRIP) phenomena are observed for the methyl group of acetate, which suggests that these groups rotate to some extent at low temperatures in DNP juice.

*Can you explain your observation "where the [proton] polarization is higher, the peak intensities become more equal"? Can you explain why "[proton] polarizations (...) were observed to decrease linearly with increasing 13C peak asymmetries"? As you wrote, this behaviour is opposite to what was observed by Mammoli et al. and by Aghelnejad et al. It seems that polarisation and asymmetry should increase together.*

The phenomenon at hand is related to the influence which the 1H polarization has on the 13C lineshape. When the 1H polarization is low a single peak is observed in the 13C NMR spectrum, which indicates a high asymmetry according to the definition given in our manuscript. In the opposite case, *i.e.,* when the 1H polarization is high two peaks are observed in the 13C NMR spectrum, which indicates a low asymmetry. Looking at the 13C NMR spectra acquired at the CP contacts in our experiment, at which point the 1H polarization has reached a constant value whilst the 13C NMR signal continues to grow, it is observed that there is minimal distortion to the 13C lineshape, indicating that it is the 1H polarization which is mostly responsible for this phenomenon. It is indeed an interesting observation, and one which goes against the rest of the literature surrounding this field. We felt that it was worth reporting, even though this phenomenon is not completely understood yet. There may also be other ways to interpret the data, but our current approach is intentionally one of the most simplistic ways we have found so far.

*What do you expect to see in the presence of "solid-state methyl group AE population imbalances at low temperatures"?*

We were simply suggesting that if a fully deuterated DNP solvent was to be used, we may be able to see changes in the 1H spectra of [2-13C]acetate as a function of the microwave irradiation period, and hence 1H polarization, which might reveal information about the build-up of AE population imbalances. The manuscript will be amended to make this point clearer.

*For practitioners of DNP, it would be nice to know more details about how you determined the "power of ca. 80 $P\mu w$ = 125 mW at the output of the microwave source and ca. $P\mu w$ = 30 mW reaching the DNP cavity (evaluated by monitoring the helium bath pressure.)" Is the first number given by the manufacturer of the microwave source? How did you convert pressure changes into mW? How do you calibrate this empirical relationship?*

125 mW is given by *VDI* (our microwave source provider). 30 mW was determined by comparison with the heating from a resistor in the bath and calibrating how much the bath pressure increases vs. power. In practice, the measurement was performed as follows:

- The variable temperature insert (VTI) was filled with liquid helium and pumped down to 0.65 mbar, corresponding to 1.2 K.
- The change of pressure when turning on a resistive heater or the microwave source for 120 s was monitored. The pressure plateaus after approximatively 60 s.
- The pressure difference between the base pressure and that under the effect of the resistive heater or the microwave irradiation $\Delta P$ is calculated.

All measurements were performed ensuring that the liquid helium level in the VTI was not varying more than a few centimetres: the microwave cavity was immersed under 5-10 cm of liquid helium. The measurements performed using the resistive heater with power $P_{heater}$ are used to plot a calibration curve $P_{heater}$ vs. $\Delta P$ with slope $a$. The deposited microwave power in the cavity is then obtained by computing $P_{microwave} = a\Delta P$. This information will be added to experiments section of the main text.

*The statement "the 1H and 13C [transverse] relaxation time constants in the presence of an rf-field are extended by orders of magnitude" seems a bit exaggerated. Bornet et al. have shown experimentally that T_1rho can be extended by a factor 5 or so when the microwaves are switched on.*

Thank you for pointing this out to us. We will make this amendment to the manuscript.

---

## Author Comment (AC2)

Review 1: Andrea Capozzi.

*Where is the asymmetry coming from? [...]*

We thank the reviewer for raising this interesting point. Given the complexity of these 13C NMR lineshapes, we decided to limit the scope of this article to reporting the observed effect, quantifying it, and linking it to the absolute polarization measured by conventional means (by comparison with thermal equilibrium NMR signal). We agree with the reviewer that more information should be given about the origin of this asymmetry, we will therefore add a paragraph listing the different interactions involved:

"The 13C NMR lineshape of [2-13C]sodium acetate has features which mainly originate from 13C chemical shift anisotropy (CSA) (typ. ~1.5 kHz at our magnetic field of 7.05 T) and 1H-13C dipolar couplings (typ. -22.7 kHz) that are affected by possible methyl group rotation. Since the 13C CSA is negligible with respect to the 1H-13C dipolar couplings, it is assumed that the 1H-13C dipolar couplings play the key role in the 13C NMR lineshape of [2-13C]sodium acetate."

*Usually, when introducing a new methodology (not yet established and broadly acknowledged), the new methodology has to the compared to the traditional one (i.e. measuring the enhancement from the ratio of the hyperpolarized proton signal to the thermal equilibrium one). In the methods part, you say to have a background free coil. Why don't you use it to show how much your new method is consistent and reliable?*

Indeed, Figure 5 shows the asymmetry vs. the absolute polarization measured by comparison with the thermal equilibrium NMR signal. We thank the reviewer for pointing this out, and we will add a sentence in the corresponding Figure caption to make this point clearer.

*Moreover, do you need a methyl group or a simple coupling with a 1H nucleus can provide the same results?*

We believe that the strong HC dipolar coupling is involved in this phenomenon.

*I would have expected at least one more probe molecule, in particular because in the discussion you mention sodium [1-13]formate that has no methyl group.*

We agree that it would be very interesting to investigate other molecules, and we intend to do so in the future. However, we have only seen such interesting lineshapes, *i.e.*, ones with similarly pronounced features, for [2-13C]acetate thus far. Other 13C-labelled methyl groups, such as gamma-picoline, and 13C-labelled methylene groups might also exhibit exotic spectra under our experimental *d*DNP conditions, and it is also possible that such effects could be observed in [1-13C]formate, if indeed one of the main requirements is the presence of a strong HC dipolar coupling. However, we have not yet investigated other suitable molecular candidates.

*Last but not least, when you calculate the asymmetry you take into account the peaks intensity. These peaks are far from being resolved and the intensity of the first will most likely be influenced by the intensity of the second and vice versa. At the end of the discussion, you mention the correct data processing procedure to estimate the asymmetry (to deconvolute the two peaks by means of Voigt fits and evaluate the integral). I agree it is less straight forward than just measuring the intensity, but you are introducing a new methodology. You should at least prove that "the peak intensity method" provides, withing a decent error margin, the same result of the "the peak fitting method". If you demonstrate that, then we will all use the intensity one, of course.*

We agree that it would be interesting to compare our peak picking method with a fitting method. However, since the lineshape is more complicated than simply two lines, we propose an alternative here, alongside our original method, which is a simple calculation of an asymmetry parameter as described below, which can be easily applied and generalized to any lineshape:

1. calculate center of gravity when $P(^1H) = 0$

$f(\omega)$

$$\int_{-\infty}^{\infty} f(\omega)\, d\omega = 1 \quad (\text{normalize})$$

$$\omega_0 = \int_{-\infty}^{\infty} \omega\, f(\omega)\, d\omega$$

2. keep $\omega_0$ constant, and calculate asymmetry with

$$A = \int_{-\infty}^{\infty} (\omega - \omega_0)\, f(\omega)\, d\omega$$

This procedure should work very easily and give a very general way to calculate asymmetry on any lineshape. In this way, any laboratory/group can adopt the procedure and reproduce the result. We furthermore build on the procedure given above and define the following quantity:

$$\delta_{\omega_0} = \frac{\int_{-\infty}^{\infty} (\omega - \omega_0(P_H = 0\%))\, f(\omega)\, d\omega}{\sqrt{\int_{-\infty}^{\infty} (\omega(P_H = 0\%) - \omega_0(P_H = 0\%))^2\, f(\omega(P_H = 0\%))\, d\omega}}$$

which is normalized by the linewidth (at FWHM at $P_H = 0\%$) to yield a dimensionless quantity. The procedure above produces the results of the kind show below, with $\delta_{\omega_0}$ being the normalized shift from $\omega_0$ at $P_H = 0\%$:

[Figure]

A line of best fit had been added in the linear regime of the curve as an additional tool for the reader. This methodology is also robust with respect to inhomogeneous magnetic fields. A discussion of the above kind will be added to the manuscript, alongside the discussion of our original approach.

*In the Methods you put a lot of emphasis on the necessity to have cross polarization to be able to use this method. Firstly, I think that with such a level of deuteration, at 6.7 T and with microwave modulation you should be able to achieve a decent SNR with direct 13C DNP as well (see T. Cheng, PCCP 2013, 15 (48)). Secondly, if CP is required, 90% of the DNP users around the world could not take advantage of this method. Lastly, in the motivation you say that "in lack of 1H rf coil" your method can be useful…How can you do CP without a proton coil???*

We agree with the reviewer that CP is not necessary. We simply use it to offer more signal-to-noise on the 13C side of our experiment. We will modify the Methods Section accordingly to stress the fact that CP is optional. Indeed, we measured the asymmetry without CP during with first data points of Figure 2.

*IMHO the bare minimum to consider this paper for publication is to address these 3 points: give some theoretical insight about the spectral features; run new experiments to compare the new way of measuring 1H polarization with the traditional way (preferably using also one more molecule); try without CP.*

As suggested by the reviewer, we give more theoretical insights although this is not within the initial scope of this article. We indicate that we compare the measured polarizations with the traditional method, and we will also make it clear that CP is an optional tool, and that the asymmetry can be measured without CP just as well.

Line 11. "is emerging" for a technique invested in 2003 is not appropriate, rephrase like "dDNP allows to prepare proton polarization…"

We will use the suggested rephrasing of the reviewer at the appropriate point in our article.

Line 33. I think there are too many references. You are mentioning hyperpolarization methodology (dDNP, PHIP, SEOP, brute force). I suggest choosing one for each, there is no need to cite 5 or 6 reviews about dDNP. Moreover, the bullet DNP technique from Benno Meier is missing, please include it.

We will make the suggested changes at the appropriate point in our article and add the missing reference to Benno Meier's work.

Line 37. Here you can cite more recent papers. A lot has been achieved from 2013 to 2021 concerning human trials. Some suggestions: Kurhanewicz, J. et al. Hyperpolarized (13)C MRI: Path to Clinical Translation in Oncology. Neoplasia **21**, 1–16 (2019); Chen, H.-Y. et al. Hyperpolarized 13C-pyruvate MRI detects real-time metabolic flux in prostate cancer metastases to bone and liver: a clinical feasibility study. Prostate Cancer Prostatic Dis. (2019) doi:10.1038/s41391-019-0180-z; Gallagher, F. A. et al. Imaging breast cancer using hyperpolarized carbon-13 MRI. Proc. Natl. Acad. Sci. **117**, 2092–2098 (2020).

We thank the reviewer for their comments on this and will include the above references.

Line 38. It is a super general statement. For sure Prof Jannin is a big player in the field, but I would cite Abragam, Goldman and or Borghini here.

We will make the suggested changes to the main text here.

Line 56. This is conclusion not introduction.

We will move this statement to the conclusion.

Line 65. You use one sample, call it just "the sample", no need to call it **I.**

We will remove all instances of "**I**" in the main text, and simply refer to the sample as suggested by the reviewer.

Line 77. Add brand and model of microwave source.

This information will be added to the manuscript at this point.

Line 91. How did you calculate 11 ms? Is there a particular reason (demanding spectrometer duty cycle)? In a solid-state saturation sequence the important thing is that the inter-pulse delay is > 3T2*. I guess your proton line is 40 – 50 kHz broad. Therefore, the T2* is 20 us. With an inter-pulse delay of 100 us the magnetization in the xy plane is completely dephased before the next pulse comes and you do not risk flipping back magnetization on z.

In our experimental setup, the above delay combined with the phase cycling given in Figure 1 of the manuscript is sufficient to remove all magnetization before commencing each experiment. Even if this procedure does not work perfectly, the amount of magnetization maintained is very small compared to what is built-up by 1H DNP or CP.

Line 95. Is the acquisition time of the FID perturbating the signal? Consider rephrasing

A suitable rephrasing will be given at the point in the manuscript.

Line 129. The mw gating paragraph is not Methods, it is discussion how it is written here.

Thank you to the reviewer for point this out. It should indeed be included in the Methods Section. We will make suitable amendments to the manuscript to move this information to the Methods section.

Line 151. This paragraph is Discussion not Results. Moreover, the explanation is not clear. Consider rephrasing like: "The microwaves are ON and the 13C nuclear ensemble relaxes towards the spin temperature value it would have achieved in the case of direct DNP (no cross polarization)". Moreover, there is a 3$^{rd}$ factor to consider. The radical concentration and temperature are in the good range for thermal mixing (Guarin et al, JPCL 2017, 8 (22) ): the 13C are polarized, the 1H are saturated. The two nuclear pools most likely exchange energy via the electron non-Zeeman reservoir. This also affects the time evolution of the 13C until 1H achieves the same spin temperature.

This text will be moved to the Discussion Section and will also be made much clearer as per the suggestion of the reviewer. We also acknowledge the important contribution from the reviewer regarding thermal mixing. We will add a discussion of this kind at the appropriate place of the main text to address this comment.

Line 173. Are not mirror images of each other with respect to the x-axis or the y-axis?

With respect to a reflection about the *y*-axis (after a 180° phase correction). This issue will be made clear at this point of the main text.

Line 196. If you use a stretched exponential, technically, you are not using "a sole" build-up time constant, but a linear combination of many (infinite) build up time constants. Moreover, in the discussion I would like to understand why you observe a stretched buildup. Is beta close to 1 or to 0.5? Please provide the value. Could it be that having polarized 13C and depolarized 1H forces a stretched exponential buildup of protons?

This is a "typo" and will be corrected in the revised version of the manuscript. Beta is close to 0.77 (+ve DNP) and 0.87 (-ve DNP). We originally decided to remove a discussion of this behaviour from the manuscript, since it is a different and complicated issue. It is possible that having highly polarized 13C nuclear spins yields a stretched exponential behaviour of the 1H polarization build-up or that the methyl group protons polarize rapidly (possibly due to enhanced nuclear spin relaxation from the adjacent $^{13}$C spin labelled site) whilst the bulk of the sample polarizes more slowly (likely attributable to $^{1}$H-$^{1}$H spin diffusion processes).

Line 236. I don't understand this sentence. Can you justify why your calibration curve (pol vs asymmetry) changes slope between positive and negative DNP?.

We believe that radiation damping is responsible for the difference between the two curves. In the case of negative DNP, radiation damping leads to an overestimation of the 1H polarization, particularly at high 1H polarizations, and hence a change in the slope of the calibration curve.

Line 239. This is discussion.

This text will be moved to the Discussion Section.

Line 256. This remains an open question until you don't measure the polarization in the traditional way.

Figure 5 shows the asymmetry vs. the absolute polarization measured by comparison with the thermal equilibrium NMR signal. We will add a sentence in the caption of Figure 5 to make this point clearer.

Line 87. "DNP equilibrium" not "DNP equilibria".

This "typo" will be corrected.

Line 87. I have never heard the term "crusher" rf pulses. Consider using "saturating" rf-pulses.

This text, and Figure 1, will be changed from "crusher" to "saturating".

---

## Author Comment (AC3)

Review 2: Anonymous.

The appearance of the spectra in Figures 3a) and 3d) are not "single peaks" but rather spectra with multiple shoulders (spanning several 100 ppm). What is the origin of those? I would be curious to know if the appearance of the spectra is identical at shorter 1H DNP times in the sequence – does this line(shape) build homogeneously?

We see at least five peaks in the 13C NMR spectrum of our sample. The peaks do not appear equally spaced, and so are not a multiplicity effect. We therefore believe that this corresponds to different chemical environments, being polarized on different time scales, which could be associated with complicated factors such as inhomogeneous sample freezing. There is a chemical environment at approximately -300ppm, which is very deshielded for a 13C nuclear spin and is perhaps an effect of hyperfine couplings with the radical spins. The 13C NMR spectra in fact change most rapidly at shorter 1H DNP times when the rate of change of the 1H polarization level is fastest. The 13C NMR lineshape does indeed also change homogenously, as shown by the 13C NMR spectra presented in the supplement. Furthermore, looking at the 13C NMR spectra acquired at the CP contacts in our experiment, at which point the 1H polarization has reached a constant value whilst the 13C NMR signal continues to grow, it is observed that there is minimal distortion to the 13C lineshape, indicating that it is the 1H polarization is mostly responsible for this phenomenon

What is the role of CSA or possible dipole-dipole interactions, and how are those manifest under both positive and negative microwave irradiation? What is the preferred energy state for coupling to P(1H) = + versus P(1H) = -?

We agree with the reviewer that more information should be given about the origin of this asymmetry, we will therefore add a paragraph listing the different interactions involved:

"The 13C NMR lineshape of [2-13C]sodium acetate has features which mainly originate from 13C CSA (typ.  $\sim$ 1.5 kHz at our magnetic field of 7.05 T) and 1H-13C dipolar couplings (typ. -22.7 kHz) that are affected by possible methyl group rotation. Since the 13C CSA is negligible with respect to the 1H-13C dipolar couplings, it is assumed that the 1H-13C dipolar couplings play the key role in the 13C NMR lineshape of [2-13C]sodium acetate."

Presumably the glycerol carbon and the quaternary carbon of the formate both contribute to the spectrum. Where are those, and how are those influenced by both cross-polarization and microwave irradiation?

The main 13C peak includes contributions from the 13C Glycerol-d8 peaks (these peaks are typically within ca. 30ppm of the [2-13C]sodium acetate peak). Although Glycerol-d8 is deuterated it can still be polarized by 1H-13C cross-polarization (CP), but such deuterated systems typically require much longer CP contact times for efficient polarization transfer. We therefore hypothesize that the influence of the Glycerol-d8 13C nuclear spins has a reduced impact on the hyperpolarized spectrum of [2-13C]sodium acetate than at lower levels of 13C polarization. Under microwave irradiation, the natural abundance 13C spins of Glycerol-d8 will be polarized by microwave irradiation with their own build-up rate and maximum polarization, although this is anticipated to be slower and lower than those of [2-13C]sodium acetate. We did not trial formate in this study.

At extended 1H DNP times, there are additional intriguing details – the claim that these are now two separate resonances doesn't quite fit with the initial picture (of a "single [peak]"). Defining Eq 1 based on the fractional intensities of these two "peaks" feels somewhat arbitrary. Without knowing what these I\_h and I\_l features represent, it's somewhat difficult to tell this is arising from the 1H polarization or from some other effect.

We propose an alternative here, alongside our original method, which is a simple calculation of an asymmetry parameter as described below, which can be easily applied and generalized to any lineshape:

This procedure should work very easily and give a very general way to calculate asymmetry on any lineshape. In this way, any laboratory/group can adopt the procedure and reproduce the result. We furthermore build on the procedure given above and define the following quantity:

$$\delta_{\omega_0} = \frac{\int_{-\infty}^{\infty} (\omega - \omega_0 (P_H = 0\%)) f(\omega) \, d\omega}{\sqrt{\int_{-\infty}^{\infty} (\omega (P_H = 0\%) - \omega_0 (P_H = 0\%))^2 f(\omega (P_H = 0\%)) \, d\omega}}$$

which is normalized by the linewidth (at FWHM at  $P_{\rm H} = 0\%$ ) to yield a dimensionless quantity. The procedure above produces the results of the kind show below, with  $\delta_{\omega_0}$  being the normalized shift from  $\omega_0$  at  $P_{\rm H} = 0\%$ :

A line of best fit had been added in the linear regime of the curve as an additional tool for the reader. This methodology is also robust with respect to inhomogeneous magnetic fields. A discussion of the above kind will be added to the manuscript, alongside the discussion of our original approach.

The different slopes for Figure 5 are explained empirically (lines 36-37) but is there a physical reason why the 1H polarization (or the asymmetry of the carbon resonances) would be more sensitive to negative microwave irradiation?

The reviewer is indeed correct and we thank them for this comment. We believe that radiation damping is responsible for the difference between the two curves. In the case of negative DNP, radiation damping leads to an overestimation of the 1H polarization, particularly at high 1H polarizations, and hence a change in the slope of the calibration curve.

Minor point: the term "crusher" is unfamiliar to me. Do you mean "saturation" sequence or "saturating comb"?

This text, and Figure 1, will be changed from "crusher" to "saturating".

---

## Author Response (AR1)

13 July 2021

Dear Editor,

Please find below our response to the reviewer comments on our article "Solid-State $^1$H Spin Polarimetry by $^{13}$CH$_3$ Nuclear Magnetic Resonance".

We are extremely grateful to the three reviewers for evaluating our manuscript very positively, and we have addressed below all the comments raised by these reviewers.

The reviewer comments are given in black, our response in blue and our amendments to the manuscript in purple.

We will submit a revised version of manuscript "mr-2021-25".

Please do not hesitate to contact me if you require any further information.

Yours,
Stuart J. Elliott.

Dr. Stuart J. Elliott
Address: Department of Chemistry, University of Liverpool, Liverpool L69 7ZD, United Kingdom
Email: Stuart.Elliott@liverpool.ac.uk
Telephone: +44 (0) 7805 888 763

Review 1: Geoffrey Bodenhausen.

*The paper on "Solid-State 1H Spin Polarimetry by 13CH3 Nuclear Magnetic Resonance" by Stuart J. Elliott, Quentin Stern (or should it be Quentin Chappuis, as in the Supplement?) and Sami Jannin raises all sorts of interesting questions.*

The surname of our colleague has been amended in the supplement. Thank you for pointing this out to us.

*When the authors write "The frequency of the dashed line corresponds to the minimum between the two peaks at high levels of 1H polarization," they discuss the two peaks phenomenologically as if they constituted a doublet. Are they implicitly referring to a doublet due to the heteronuclear dipole-dipole coupling in the Hamiltonian?*

We believe that the strong HC dipolar coupling is involved in this phenomenon. We agree with the reviewer that more information should be given about the origin of this asymmetry. We have therefore added the below paragraph listing the different interactions involved to the manuscript:

The $^{13}$C NMR lineshapes of [2-$^{13}$C]sodium acetate shown in Figure 3 have features which mainly originate from $^{13}$C chemical shift anisotropy (CSA) (max. ~1.5 kHz at our magnetic field of 7.05 T) and $^1$H-$^{13}$C dipolar couplings (typ. -22.7 kHz) that are affected by possible methyl group rotation. Since the $^{13}$C CSA is negligible with respect to the $^1$H-$^{13}$C dipolar couplings, it is assumed that the $^1$H-$^{13}$C dipolar couplings play the key role in the $^{13}$C NMR lineshape of [2-$^{13}$C]sodium acetate.

*Is this splitting only seen in methyl groups? Do these methyl groups have to rotate freely all the way down to sample temperatures near 1.2 K? Is that the case for methyl groups in acetate? Do you see similar effects in molecules such as gamma picoline that are known to have very low rotation barriers? If the methyl rotation in acetate is frozen, why are the methyl signals in acetate any different from those of other rigid chemical moieties like CH2 groups? You suggest that similar effects can be observed in molecules "such as [1- 13C]sodium formate", although there are no methyl groups in formate.*

We have only seen such interesting lineshapes, *i.e.*, ones with similarly pronounced features, for [2-13C]acetate thus far. Other 13C-labelled methyl groups, such as gamma-picoline, and 13C-labelled methylene groups might also exhibit exotic spectra under our experimental *d*DNP conditions, and it is also possible that such effects could be observed in [1-13C]formate, if indeed one of the main requirements is the presence of a strong HC dipolar coupling. However, we have not yet investigated other suitable molecular candidates. To our knowledge, the observed asymmetry does not require methyl group rotation, and only a strong HC dipolar coupling. Particularly fast methyl group rotation is likely to average HC dipolar couplings and is likely to work against the phenomena observed here. As can be seen from J.-N. Dumez et al., *J. Phys. Chem. Lett.* **2017**, 8, 3549-3555, quantum rotor induced polarization (QRIP) phenomena are observed for the methyl group of acetate, which suggests that these groups rotate to some extent at low temperatures in DNP juice.

*Can you explain your observation "where the [proton] polarization is higher, the peak intensities become more equal"? Can you explain why "[proton] polarizations (...) were observed to decrease linearly with increasing 13C peak asymmetries"? As you wrote, this behaviour is opposite to what was observed by Mammoli et al. and by Aghelnejad et al. It seems that polarisation and asymmetry should increase together.*

The phenomenon at hand is related to the influence which the 1H polarization has on the 13C lineshape. When the 1H polarization is low a single peak is observed in the 13C NMR spectrum, which indicates a high asymmetry according to the definition given in our manuscript. In the opposite case, *i.e.,* when the 1H polarization is high, two peaks are observed in the 13C NMR spectrum which indicates a low asymmetry. Looking at the 13C NMR spectra acquired at the CP contacts in our experiment, at which point the 1H polarization has reached a constant value whilst the 13C NMR signal continues to grow, it is observed that there is minimal distortion to the 13C lineshape, indicating that it is the 1H polarization which is mostly responsible for this phenomenon. It is indeed an interesting observation, and one which goes against the rest of the literature surrounding this field. We felt that it was worth reporting, even though this phenomenon is not completely understood yet. There may also be other ways to interpret the data, but our current approach is intentionally one of the most simplistic ways we have found so far.

*What do you expect to see in the presence of "solid-state methyl group AE population imbalances at low temperatures"?*

We were simply suggesting that if a fully deuterated DNP solvent was to be used, we may be able to see changes in the 1H spectra of [2-13C]acetate as a function of the microwave irradiation period, and hence 1H polarization, which might reveal information about the build-up of AE population imbalances.

*For practitioners of DNP, it would be nice to know more details about how you determined the "power of ca. 80 $P_{\mu w}$ = 125 mW at the output of the microwave source and ca. $P_{\mu w}$ = 30 mW reaching the DNP cavity (evaluated by monitoring the helium bath pressure.)" Is the first number given by the manufacturer of the microwave source? How did you convert pressure changes into mW? How do you calibrate this empirical relationship?*

**2.4. Microwave Power Evaluation**

The microwave power reaching the DNP cavity was determined by comparison with the heating from a resistor in the liquid helium bath and calibrating how much the bath pressure increases vs. microwave power. In practice, the measurement was performed as follows:

   (*i*) The VTI was filled with liquid helium and pumped down to 0.65 mbar, corresponding to 1.2 K;

   (*ii*) The change of pressure when turning on a resistive heater or the microwave source for 120 s was monitored. The pressure plateaus after approximatively 60 s;

   (*iii*) The pressure difference between the base pressure and that under the effect of the resistive heater or the microwave source $\Delta P_{mbar}$ is calculated.

   All measurements were performed ensuring that the liquid helium level in the VTI was not varying by more than a few centimetres: the microwave cavity was immersed under 5-10 cm of liquid helium. The measurements performed using the resistive heater with power $P_{heater}$ are used to plot a calibration curve $P_{heater}$ vs. $\Delta P_{mbar}$ with slope $a$. The deposited microwave power in the cavity is then obtained by computing $P_{microwave} = a\Delta P_{mbar}$.

[Figure]

*The statement "the 1H and 13C [transverse] relaxation time constants in the presence of an rf-field are extended by orders of magnitude" seems a bit exaggerated. Bornet et al. have shown experimentally that T_1rho can be extended by a factor 5 or so when the microwaves are switched on.*

For 1H spins, the above relaxation time constants are really extended by orders of magnitude (see: *Phys Chem Chem Phys* **2016**, 18, 30530-30535).

Reviewer 2: Andrea Capozzi.

*Where is the asymmetry coming from? […]*

We thank the reviewer for raising this interesting point. Given the complexity of these 13C NMR lineshapes, we decided to limit the scope of this article to reporting the observed effect, quantifying it, and linking it to the absolute 1H polarization measured by conventional means (by comparison with thermal equilibrium NMR signal). We agree with the reviewer that more information should be given about the origin of this asymmetry. We have therefore added the below paragraph listing the different interactions involved to the manuscript:

The 13C NMR lineshapes of [2-13C]sodium acetate shown in Figure 3 have features which mainly originate from 13C chemical shift anisotropy (CSA) (max. ~1.5 kHz at our magnetic field of 7.05 T) and 1H-13C dipolar couplings (typ. -22.7 kHz) that are affected by possible methyl group rotation. Since the 13C CSA is negligible with respect to the 1H-13C dipolar couplings, it is assumed that the 1H-13C dipolar couplings play the key role in the 13C NMR lineshape of [2-13C]sodium acetate.

*Usually, when introducing a new methodology (not yet established and broadly acknowledged), the new methodology has to the compared to the traditional one (i.e. measuring the enhancement from the ratio of the hyperpolarized proton signal to the thermal equilibrium one). In the methods part, you say to have a background free coil. Why don't you use it to show how much your new method is consistent and reliable?*

Indeed, Figure 5 shows the asymmetry vs. the absolute polarization measured by comparison with the thermal equilibrium NMR signal. We thank the reviewer for pointing this out, and we have added the below sentence in the caption of Figure 5 to make this point clearer:

The absolute 1H polarizations $P_H$ were measured by comparison with a thermal equilibrium 1H NMR signal.

*Moreover, do you need a methyl group or a simple coupling with a 1H nucleus can provide the same results?*

We believe that the strong HC dipolar coupling is involved in this phenomenon.

*I would have expected at least one more probe molecule, in particular because in the discussion you mention sodium [1-13]formate that has no methyl group.*

We agree that it would be very interesting to investigate other molecules, and we intend to do so in the future. However, we have only seen such interesting lineshapes, *i.e.*, ones with similarly pronounced features, for [2-13C]acetate thus far. Other 13C-labelled methyl groups, such as gamma-picoline, and 13C-labelled methylene groups might also exhibit exotic spectra under our experimental *d*DNP conditions, and it is also possible that such effects could be observed in [1-13C]formate, if indeed one of the main requirements is the presence of a strong HC dipolar coupling. However, we have not yet investigated other suitable molecular candidates.

*Last but not least, when you calculate the asymmetry you take into account the peaks intensity. These peaks are far from being resolved and the intensity of the first will most likely be influenced by the intensity of the second and vice versa. At the end of the discussion, you mention the correct data processing procedure to estimate the asymmetry (to deconvolute the two peaks by means of Voigt fits and evaluate the integral). I agree it is less straight forward than just measuring the intensity, but you*

*are introducing a new methodology. You should at least prove that "the peak intensity method" provides, withing a decent error margin, the same result of the "the peak fitting method". If you demonstrate that, then we will all use the intensity one, of course.*

We agree that it would be interesting to compare our peak picking method with a fitting method. However, since the lineshape is more complicated than simply two lines, we have proposed an alternative here which is a simple calculation of a parameter that can be easily applied and generalized to any lineshape. This methodology is also robust with respect to inhomogeneous magnetic fields and should work very easily and give a very general way to calculate asymmetry on any lineshape. In this way, any laboratory can adopt the procedure and reproduce the result.

The $^{13}C$ NMR lineshapes presented in Figure 3 are complicated and so it is desirable to construct a parameter which can describe the $^{1}H$ polarization $P_H$, be robust with respect to field inhomogeneities and easily applied to any lineshape. Figure 4 therefore also displays the $^{13}C$ NMR peak CoG deviation $\delta_{\omega_0}$ as a function of the $^{1}H$ DNP time for the case of positive microwave irradiation. The $^{13}C$ NMR peak CoG normalized deviation $\delta_{\omega_0}$ is defined as:

$$\delta_{\omega_0} = \frac{M_{asym}}{LW_0} \quad (1)$$

where $M_{asym}$ is denoted as the first moment of asymmetry and corresponds to the following quantity:

$$M_{asym} = \int_{-\infty}^{\infty} (\omega - \omega_0(P_H = 0\%)) f(\omega) \, d\omega \quad (2)$$

The first moment of asymmetry $M_{asym}$ is based on a calculation whereby the CoG of the $^{13}C$ NMR peak $\omega_0$ is held constant at $\omega_0(P_H = 0\%)$, *i.e.*, the $^{13}C$ NMR peak CoG corresponding to when the $^{1}H$ polarization $P_H$ is zero. The CoG of the $^{13}C$ NMR peak $\omega_0$ is calculated as:

$$\omega_0 = \int_{-\infty}^{\infty} \omega f(\omega) \, d\omega \quad (3)$$

where the intensities of the $^{13}C$ NMR peaks are normalized:

$$\int_{-\infty}^{\infty} f(\omega) \, d\omega = 1 \quad (4)$$

where $\omega$ is the resonance frequency and $f(\omega)$ is the peak intensity at $\omega$. The procedure outlined above ensures that $M_{asym} = 0$ at $P_H = 0\%$ such that the described approach can be readily generalized to any lineshape. The quantity $LW_0$ is a measure of the linewidth of the $^{13}C$ NMR peak in the case of $P_H = 0\%$:

$$LW_0 = \sqrt{\int_{-\infty}^{\infty} (\omega(P_H = 0\%) - \omega_0(P_H = 0\%))^2 f(\omega(P_H = 0\%)) \, d\omega} \quad (5)$$

*i.e.*, the square root of the second moment at $P_H = 0\%$. This factor establishes a $^{13}C$ NMR peak CoG deviation $\delta_{\omega_0}$ (defined in Equation 1) which is a normalized and dimensionless quantity.

The procedure above produces results of the kind shown below:

[Figure]

[Figure]

**Figure 4: Experimental ¹H polarization $P_H$ DNP build-up curve (black filled squares and left-hand axis) and ¹³C NMR peak CoG normalized deviation $\delta_{\omega_0}$ (grey empty circles and right-hand axis) as a function of the ¹H DNP time acquired at 7.05 T (¹H nuclear Larmor frequency = 300.13 MHz, ¹³C nuclear Larmor frequency = 75.47 MHz) and 1.2 K with a single transient per data point for the case of positive microwave irradiation. The timings coincide with those shown in Figure 2. The black solid line indicates the best fit of the experimental data points for the ¹H polarization $P_H$ DNP build-up curve, and has the corresponding fitting function: $A(1-\exp\{-(t/\tau_{DNP}^{\pm})^{\beta}\})$. Mean ¹H DNP build-up time constant: $\langle\tau_{DNP}^{+}\rangle$ = 80.2 ± 0.3 s.**

[Figure]

**Figure 5: Experimental ¹H polarizations $P_H$ as a function of the ¹³C NMR peak CoG normalized deviation $\delta_{\omega_0}$ acquired at 7.05 T (¹H nuclear Larmor frequency = 300.13 MHz, ¹³C nuclear Larmor frequency = 75.47 MHz) and 1.2 K with a single transient per data point for the case of positive microwave irradiation. The experimental data were fitted with a phenomenological function: $P_H(\delta_{\omega_0}) = A \times \delta_{\omega_0}^{\beta}$. Best fit values: $A$ = 129.1% ± 0.8%; $\beta$ = 0.736 ± 0.005. The absolute ¹H polarizations $P_H$ were measured by comparison with a thermal equilibrium ¹H NMR signal.**

*In the Methods you put a lot of emphasis on the necessity to have cross polarization to be able to use this method. Firstly, I think that with such a level of deuteration, at 6.7 T and with microwave modulation you should be able to achieve a decent SNR with direct 13C DNP as well (see T. Cheng, PCCP 2013, 15 (48)). Secondly, if CP is required, 90% of the DNP users around the world could not take advantage of this method. Lastly, in the motivation you say that "in lack of 1H rf coil" your method can be useful…How can you do CP without a proton coil???*

We agree with the reviewer that CP is not necessary. We simply use it to offer more signal-to-noise on the 13C side of our experiment. We have modified the Methods Section accordingly to stress the fact that CP is optional. Indeed, we measured the asymmetry without CP during with first data points of Figure 2.

It should be stressed that the use of CP is purely optional, and in most cases its use will be dictated by the *rf*-hardware available. We use CP here simply as a means to offer greater SNRs for ¹³C NMR signal detection. Given the level of sample deuteration, at 6.7 T and with microwave modulation suitable SNRs can also be achieved with direct ¹³C DNP (Chen et al., 2013).

*IMHO the bare minimum to consider this paper for publication is to address these 3 points: give some theoretical insight about the spectral features; run new experiments to compare the new way of measuring 1H polarization with the traditional way (preferably using also one more molecule); try without CP.*

As suggested by the reviewer, we have given more theoretical insights although this is not within the initial scope of this article. We have indicated that we compare the measured polarizations with the traditional method, and we have also made it clear that CP is an optional tool, and that the asymmetry can be measured without CP just as well.

Line 11. "is emerging" for a technique invested in 2003 is not appropriate, rephrase like "dDNP allows to prepare proton polarization…"

We have used the suggested rephrasing of the reviewer at the appropriate point in our article.

Dissolution-dynamic nuclear polarization is used to prepare proton polarizations approaching unity.

Line 33. I think there are too many references. You are mentioning hyperpolarization methodology (dDNP, PHIP, SEOP, brute force). I suggest choosing one for each, there is no need to cite 5 or 6 reviews about dDNP. Moreover, the bullet DNP technique from Benno Meier is missing, please include it.

We have made the suggested changes at the appropriate point in our article and we have also added the missing reference to Benno Meier's work.

Ardenkjær-Larsen, J.-H., Fridlund, B., Gram, A., Hansson, G., Hansson, L., Lerche, M. H., Servin, R., Thaning, M., and Golman, K.: Increase in signal-to-noise ratio of > 10,000 times in liquid-state NMR, Proc. Natl. Acad. Sci. U.S.A., 100, 10158-10163, https://doi.org/10.1073/pnas.1733835100, 2003.
Hirsch, M. L., Kalechofsky, N., Belzer, A., Rosay, M., and Kempf, J. G.: Brute-Force Hyperpolarization for NMR and MRI, J. Am. Chem. Soc., 137, 8428-8434, https://doi.org/10.1021/jacs.5b01252, 2015.
Dale, M. W., and Wedge, C. J.: Optically generated hyperpolarization for sensitivity enhancement in solution-state NMR spectroscopy, Chem. Commun., 52, 13221-13224, https://doi.org/10.1039/C6CC06651H, 2016.
Meier, B.: Quantum-rotor-induced polarization, Magn. Reson. Chem., 56, 610-618, https://doi.org/10.1002/mrc.4725, 2018.
Kouřil, K., Kouřilová, H., Bartram, S., Levitt, M. H., and Meier, B.: Scalable dissolution-dynamic nuclear polarization with rapid transfer of a polarized solid, Nat. Commun., 10, 1733, https://doi.org/10.1038/s41467-019-09726-5, 2019.

Line 37. Here you can cite more recent papers. A lot has been achieved from 2013 to 2021 concerning human trials. Some suggestions: Kurhanewicz, J. et al. Hyperpolarized (13)C MRI: Path to Clinical Translation in Oncology. Neoplasia **21**, 1–16 (2019); Chen, H.-Y. et al. Hyperpolarized 13C-pyruvate MRI detects real-time metabolic flux in prostate cancer metastases to bone and liver: a clinical feasibility study. Prostate Cancer Prostatic Dis. (2019) doi:10.1038/s41391-019-0180-z; Gallagher, F. A. et al. Imaging breast cancer using hyperpolarized carbon-13 MRI. Proc. Natl. Acad. Sci. **117**, 2092–2098 (2020).

We thank the reviewer for their comments on this and we have now included the below references:

Nelson, S. J., Kurhanewicz, J., Vigneron, D. B., Larson, P. E. Z., Harzstark, A. L., Ferrone, M., van Criekinge, M., Chang, J. W., Bok, R., Park, I., Reed, G., Carvajal, L., Small, E. J., Munster, P., Weinberg, V. K., Ardenkjær-Larsen, J.-H., Chen, A. P., Hurd, R. E., Odegardstuen, L.-I., Robb, F. J., Tropp, J., and Murray, J. A.: Metabolic imaging of patients with prostate cancer using hyperpolarized [1-$^{13}$C]pyruvate, Sci. Trans. Med., 5, 198ra108, https://doi.org/10.1126/scitranslmed.3006070, 2013.
Chen, H.-Y., Aggarwal, R., Bok, R. A., Ohliger, M. A., Zhu, Z., Lee, P., Goodman, J. W., van Criekinge, M., Carvajal, L., Slater, J. B., Larson, P. E. Z., Small, E. J., Kurhanewicz, J. and Vigeron, D. B.: Hyperpolarized $^{13}$C-pyruvate MRI detects real-time metabolic flux in prostate cancer

metastases to bone and liver: a clinical feasibility study, Prostate Cancer Prostatic Dis., 23, 269-276, https://doi.org/10.1038/s41391-019-0180-z, 2020.

Gallagher, F. A., Woitek, R., McLean, M. A., Gill, A. B., Garcia, R. M., Provenzano, E., Reimer, F., Kaggie, J., Chhabra, A., Ursprung, S., Grist, J. T., Daniels, C. J., Zaccagna, F., Laurent, M.-C., Locke, M., Hilborne, S., Frary, A., Torheim, T., Boursnell, C., Schiller, A., Patterson, I., Slough, R., Carmo, B., Kane, J., Biggs, H., Harrison, E., Deen, S. S., Patterson, A., Lanz, T., Kingsbury, Z., Ross, M., Basu, B., Baird, R., Lomas, D. J., Sala, E., Watson, J., Rueda, O. M., Chin, S.-P., Wilkinson, I. B., Graves, M. J., Abraham, J. E., Gilbert, F. J., Caidas, C., and Brindle, K. M.: Imaging breast cancer using hyperpolarized carbon-13 MRI, Proc. Natl. Acad. Sci., 117, 2092-2098, https://doi.org/10.1073/pnas.1913841117, 2020.

Line 38. It is a super general statement. For sure Prof Jannin is a big player in the field, but I would cite Abragam, Goldman and or Borghini here.

We have made the suggested changes to the main text here.

Abragam, A., and Goldman, M.: Principles of dynamic nuclear polarisation., Rep. Prog. Phys., 41, 395-467, https://doi.org/10.1088/0034-4885/41/3/002, 1978.

Line 56. This is conclusion not introduction.

We have moved this statement to the conclusion.

Line 65. You use one sample, call it just "the sample", no need to call it **I.**

We have removed all instances of "**I**" in the main text, and simply refer to the sample as suggested by the reviewer. In three instances, the sample is referred to as "[2-13C]sodium acetate".

Line 77. Add brand and model of microwave source.

This information has been added to the manuscript at this point.

"…value given by the provider of our microwave source *VDI/AMC 705…"*

Line 91. How did you calculate 11 ms? Is there a particular reason (demanding spectrometer duty cycle)? In a solid-state saturation sequence the important thing is that the inter-pulse delay is > 3T2*. I guess your proton line is 40 – 50 kHz broad. Therefore, the T2* is 20 us. With an inter-pulse delay of 100 us the magnetization in the x-y plane is completely dephased before the next pulse comes and you do not risk flipping back magnetization on z.

In our experimental setup, the above delay combined with the phase cycling described in the caption of Figure 1 is sufficient to remove all magnetization before commencing each experiment. Even if this procedure does not work perfectly, the amount of magnetization maintained is very small compared to what is built-up by 1H DNP or CP.

Line 95. Is the acquisition time of the FID perturbating the signal? Consider rephrasing

A suitable rephrasing has been given at the point in the manuscript.

The $^{13}$C Zeeman magnetization trajectory is minimally perturbed by the application of a small flip-angle *rf*-pulse (typ. $\beta$ = 3.5°) used for detection, which is then followed by a short acquisition period (typ. $t_{FID}$ = 1 ms).

Line 129. The mw gating paragraph is not Methods, it is discussion how it is written here.

We prefer the description of microwave gating as it is currently given in the manuscript.

Line 151. This paragraph is Discussion not Results. Moreover, the explanation is not clear. Consider rephrasing like: "The microwaves are ON and the 13C nuclear ensemble relaxes towards the spin temperature value it would have achieved in the case of direct DNP (no cross polarization)". Moreover, there is a 3rd factor to consider. The radical concentration and temperature are in the good range for thermal mixing (Guarin et al, JPCL 2017, 8 (22) ): the 13C are polarized, the 1H are saturated. The two nuclear pools most likely exchange energy via the electron non-Zeeman reservoir. This also affects the time evolution of the 13C until 1H achieves the same spin temperature.

This text has been moved to the Discussion Section and has also been made much clearer as per the suggestion of the reviewer. We also acknowledge the important contribution from the reviewer regarding thermal mixing and we have added a discussion of this kind at the appropriate place of the main text to address this comment.

The decay of $^{13}$C polarization during the $^{1}$H DNP build-up interval $t_{\mathrm{DNP}}^{2}$ shown in Figure 2 occurs when the microwave source is active and the $^{13}$C nuclear spin ensemble relaxes towards the spin temperature it would have achieved in the case of direct $^{13}$C DNP, *i.e.*, no CP. This $^{13}$C polarization decay is a combination of three factors: (*i*) the microwaves are active and hence polarization is diminishing towards the low DNP equilibrium of the $^{13}$C nuclear spins with TEMPOL as the polarizing agent; and (*ii*) the $^{13}$C nuclear spins are being actively pulsed, although minimally, every 5 s, which leads to an accumulative loss of $^{13}$C NMR signal intensity over many minutes; and (*iii*) the radical concentration and temperature are in an optimal range for thermal mixing (Guarin et al, 2017) and since the $^{13}$C spins are polarized whilst the $^{1}$H spins are saturated the two nuclear pools most likely exchange energy via the electron non-Zeeman reservoir, which influences the time evolution of the $^{13}$C magnetization until the $^{1}$H spins achieve the same spin temperature. The difference in the $^{13}$C polarizations $P_{\mathrm{C}}$ at $^{1}$H DNP time = 24 mins for positive and negative microwave irradiation is associated with the $^{1}$H polarization build-ups and the performance efficiency of the multiple-contact CP *rf*-pulses, see the Supplement.

Line 173. Are not mirror images of each other with respect to the x-axis or the y-axis?

With respect to a reflection about the *y*-axis (after a 180° phase correction). This statement has been made more generally at this point of the main text.

It is interesting to note that the $^{13}$C NMR spectra acquired in the cases of positive (Figure 3b) and negative (Figure 3c) microwave irradiation do not have the same overall profile at long $^{1}$H DNP times.

Line 196. If you use a stretched exponential, technically, you are not using "a sole" build-up time constant, but a linear combination of many (infinite) build up time constants. Moreover, in the discussion I would like to understand why you observe a stretched buildup. Is beta close to 1 or to 0.5? Please provide the value. Could it be that having polarized 13C and depolarized 1H forces a stretched exponential buildup of protons?

We thank the reviewer for pointing this out to us. This is a "typo" has now been corrected in the revised version of the manuscript. Beta is close to 0.77 (+ve DNP) and 0.87 (-ve DNP). We originally decided to remove a discussion of this behaviour from the manuscript, since it is a different and complicated issue. We have chosen not to address this topic again here. However, it is certainly possible that having highly polarized 13C nuclear spins yields a stretched exponential behaviour of the 1H polarization build-up or that the methyl group protons polarize rapidly (possibly due to enhanced nuclear spin relaxation from the adjacent $^{13}$C spin labelled site) whilst the bulk of the sample polarizes more slowly (likely attributable to $^{1}$H-$^{1}$H spin diffusion processes).

Line 236. I don't understand this sentence. Can you justify why your calibration curve (pol vs asymmetry) changes slope between positive and negative DNP?.

We believe that radiation damping is responsible for the difference between the two curves. In the case of negative DNP, radiation damping leads to an overestimation of the 1H polarization, particularly at high 1H polarizations, and hence a change in the slope of the calibration curve.

Line 239. This is discussion.

This text has now been moved to the Discussion Section.

Line 256. This remains an open question until you don't measure the polarization in the traditional way.

Figure 5 shows the asymmetry vs. the absolute polarization measured by comparison with the thermal equilibrium NMR signal. We have added a sentence in the caption of Figure 4 to make this point clearer.

The absolute $^1$H polarizations $P_\text{H}$ were measured by comparison with a thermal equilibrium $^1$H NMR signal.

Line 87. "DNP equilibrium" not "DNP equilibria".

This "typo" has been corrected.

Line 87. I have never heard the term "crusher" rf pulses. Consider using "saturating" rf-pulses.

This text, and Figure 1, has been changed from "crusher" to "saturating".

Reviewer 3: Anonymous.

The appearance of the spectra in Figures 3a) and 3d) are not "single peaks" but rather spectra with multiple shoulders (spanning several 100 ppm). What is the origin of those? I would be curious to know if the appearance of the spectra is identical at shorter 1H DNP times in the sequence – does this line(shape) build homogeneously?

We see at least five peaks in the 13C NMR spectrum of our sample. The peaks do not appear equally spaced, and so are not a multiplicity effect. We therefore believe that this corresponds to different chemical environments, being polarized on different time scales, which could be associated with complicated factors such as inhomogeneous sample freezing. There is a chemical environment at approximately -300ppm, which is very deshielded for a 13C nuclear spin and is perhaps an effect of hyperfine couplings with the radical spins. The 13C NMR spectra in fact change most rapidly at shorter 1H DNP times when the rate of change of the 1H polarization level is fastest. The appearance of the 13C NMR lineshapes are not identical at shorter 1H DNP times, as shown by the 13C NMR spectra presented in the supplement, which lead to our investigation of this phenomena. Furthermore, looking at the 13C NMR spectra acquired at the CP contacts in our experiment, at which point the 1H polarization has reached a constant value whilst the 13C NMR signal continues to grow, it is observed that there is minimal distortion to the 13C lineshape, indicating that it is the 1H polarization is mostly responsible for this phenomenon

What is the role of CSA or possible dipole-dipole interactions, and how are those manifest under both positive and negative microwave irradiation? What is the preferred energy state for coupling to P(1H) = + versus P(1H) = - ?

We agree with the reviewer that more information should be given about the origin of this asymmetry. We have therefore added the below paragraphs listing the different interactions involved to the manuscript:

The $^{13}$C NMR lineshapes of [2-$^{13}$C]sodium acetate shown in Figure 3 have features which mainly originate from $^{13}$C chemical shift anisotropy (CSA) (max. ~1.5 kHz at our magnetic field of 7.05 T) and $^1$H-$^{13}$C dipolar couplings (typ. -22.7 kHz) that are affected by possible methyl group rotation. Since the $^{13}$C CSA is negligible with respect to the $^1$H-$^{13}$C dipolar couplings, it is assumed that the $^1$H-$^{13}$C dipolar couplings play the key role in the $^{13}$C NMR lineshape of [2-$^{13}$C]sodium acetate.

It is also possible that the dipolar couplings and CSA interactions manifest differently under positive and negative microwave irradiation, and that there is a preferred energy state for coupling to positive and negative $^1$H polarizations $P_\text{H}$ leading to non-identical $^{13}$C NMR spectra.

[Figure]

Presumably the glycerol carbon and the quaternary carbon of the formate both contribute to the spectrum. Where are those, and how are those influenced by both cross-polarization and microwave irradiation?

The main 13C peak includes contributions from the 13C Glycerol-d8 peaks (these peaks are typically within ca. 30ppm of the [2-13C]sodium acetate peak). Although Glycerol-d8 is deuterated it can still be polarized by 1H-13C cross-polarization (CP), but such deuterated systems typically require much longer CP contact times for efficient polarization transfer. We therefore hypothesize that the influence of the Glycerol-d8 13C nuclear spins has a reduced impact on the hyperpolarized spectrum of [2-13C]sodium acetate than at lower levels of 13C polarization. Under microwave irradiation, the natural abundance 13C spins of Glycerol-d8 will be polarized by microwave irradiation with their own build-up rate and maximum polarization, although this is anticipated to be slower and lower than those of [2-13C]sodium acetate. We did not trial formate in this study.

Under microwave irradiation, the natural abundance $^{13}$C spins of glycerol-$d_8$ will be polarized with their own build-up rate and maximum polarization, and although deuterated glycerol-$d_8$ can also be polarized by $^1$H-$^{13}$C CP (Vuichoud et al, 2019).

At extended 1H DNP times, there are additional intriguing details – the claim that these are now two separate resonances doesn't quite fit with the initial picture (of a "single [peak]"). Defining Eq 1 based on the fractional intensities of these two "peaks" feels somewhat arbitrary. Without knowing what these I_h and I_l features represent, it's somewhat difficult to tell this is arising from the 1H polarization or from some other effect.

We have proposed an alternative here which is a simple calculation of a parameter that can be easily applied and generalized to any lineshape. This methodology is also robust with respect to inhomogeneous magnetic fields and should work very easily and give a very general way to calculate asymmetry on any lineshape. In this way, any laboratory can adopt the procedure and reproduce the result.

The $^{13}$C NMR lineshapes presented in Figure 3 are complicated and so it is desirable to construct a parameter which can describe the $^1$H polarization $P_H$, be robust with respect to field inhomogeneities and easily applied to any lineshape. Figure 4 therefore also displays the $^{13}$C NMR peak CoG deviation $\delta_{\omega_0}$ as a function of the $^1$H DNP time for the case of positive microwave irradiation. The $^{13}$C NMR peak CoG normalized deviation $\delta_{\omega_0}$ is defined as:

$$\delta_{\omega_0} = \frac{M_{asym}}{LW_0} \quad (1)$$

where $M_{asym}$ is denoted as the first moment of asymmetry and corresponds to the following quantity:

$$M_{asym} = \int_{-\infty}^{\infty} (\omega - \omega_0(P_H = 0\%)) \, f(\omega) \, d\omega \quad (2)$$

The first moment of asymmetry $M_{asym}$ is based on a calculation whereby the CoG of the $^{13}$C NMR peak $\omega_0$ is held constant at $\omega_0(P_H = 0\%)$, i.e., the $^{13}$C NMR peak CoG corresponding to when the $^1$H polarization $P_H$ is zero. The CoG of the $^{13}$C NMR peak $\omega_0$ is calculated as:

$$\omega_0 = \int_{-\infty}^{\infty} \omega \, f(\omega) \, d\omega \quad (3)$$

where the intensities of the $^{13}$C NMR peaks are normalized:

$$\int_{-\infty}^{\infty} f(\omega) \, d\omega = 1 \quad (4)$$

where $\omega$ is the resonance frequency and $f(\omega)$ is the peak intensity at $\omega$. The procedure outlined above ensures that $M_{asym} = 0$ at $P_H = 0\%$ such that the described approach can be readily generalized to any lineshape. The quantity $LW_0$ is a measure of the linewidth of the $^{13}$C NMR peak in the case of $P_H = 0\%$:

$$LW_0 = \sqrt{\int_{-\infty}^{\infty}(\omega(P_H = 0\%) - \omega_0(P_H = 0\%))^2 \, f(\omega(P_H = 0\%)) \, d\omega} \quad (5)$$

*i.e.*, the square root of the second moment at $P_H = 0\%$. This factor establishes a $^{13}C$ NMR peak CoG deviation $\delta_{\omega_0}$ (defined in Equation 1) which is a normalized and dimensionless quantity.

The procedure above produces results of the kind shown below:

[Figure]

**Figure 4: Experimental $^1H$ polarization $P_H$ DNP build-up curve (black filled squares and left-hand axis) and $^{13}C$ NMR peak CoG normalized deviation $\delta_{\omega_0}$ (grey empty circles and right-hand axis) as a function of the $^1H$ DNP time acquired at 7.05 T ($^1H$ nuclear Larmor frequency = 300.13 MHz, $^{13}C$ nuclear Larmor frequency = 75.47 MHz) and 1.2 K with a single transient per data point for the case of positive microwave irradiation. The timings coincide with those shown in Figure 2. The black solid line indicates the best fit of the experimental data points for the $^1H$ polarization $P_H$ DNP build-up curve, and has the corresponding fitting function: $A(1-\exp\{-(t/\tau_{DNP}^{\pm})^{\beta}\})$. Mean $^1H$ DNP build-up time constant: $\langle\tau_{DNP}^+\rangle$ = 80.2 ± 0.3 s.**

[Figure]

**Figure 5: Experimental $^1H$ polarizations $P_H$ as a function of the $^{13}C$ NMR peak CoG normalized deviation $\delta_{\omega_0}$ acquired at 7.05 T ($^1H$ nuclear Larmor frequency = 300.13 MHz, $^{13}C$ nuclear Larmor frequency = 75.47 MHz) and 1.2 K with a single transient per data point for the case of positive microwave irradiation. The experimental data were fitted with a phenomenological function: $P_H(\delta_{\omega_0}) = A \times \delta_{\omega_0}^{\beta}$. Best fit values: $A$ = 129.1% ± 0.8%; $\beta$ = 0.736 ± 0.005. The absolute $^1H$ polarizations $P_H$ were measured by comparison with a thermal equilibrium $^1H$ NMR signal.**

The different slopes for Figure 5 are explained empirically (lines 36-37) but is there a physical reason why the 1H polarization (or the asymmetry of the carbon resonances) would be more sensitive to negative microwave irradiation?

The reviewer is indeed correct, and we thank them for this comment. We believe that radiation damping is responsible for the difference between the two curves. In the case of negative DNP, radiation damping leads to an overestimation of the 1H polarization, particularly at high 1H polarizations, and hence a change in the slope of the calibration curve.

[Figure]

Minor point: the term "crusher" is unfamiliar to me.  Do you mean "saturation" sequence or "saturating comb"?

This text, and Figure 1, have been changed from "crusher" to "saturating".

---

## Author Response (AR2)

26 July 2021

Dear Editor,

Please find below our response to the editor comments on our article "Solid-State $^1$H Spin Polarimetry by $^{13}$CH$_3$ Nuclear Magnetic Resonance".

We are extremely grateful to the editor for evaluating our manuscript very positively, and we have addressed below all the comments raised by the editor.

The editor comments are given in black, our response in blue and our amendments to the manuscript in purple.

We will submit a revised version of manuscript "mr-2021-25".

Please do not hesitate to contact me if you require any further information.

Yours,
Stuart J. Elliott.

Dr. Stuart J. Elliott
Address: Department of Chemistry, University of Liverpool, Liverpool L69 7ZD, United Kingdom
Email: Stuart.Elliott@liverpool.ac.uk
Telephone: +44 (0) 7805 888 763

Editor Comments: Daniel Abergel.

Thank you for submitting your manuscript to Magnetic Resonance.

We are delighted to contribute to Geoffrey Bodenhausen's Festschrift.

You have addressed the issues raised by the reviewers, and I will be glad to accept your manuscript for publication.

We thank the editor for this positive outcome.

The sentence in line 280: "Once the $^{13}$C NMR peak CoG normalized deviation $\delta_{\omega_0}$ falls below zero the $^1$H polarization $P_H$, rapidly drops towards negative $^{13}$C NMR peak CoG normalized deviations $\delta_{\omega_0}$ (with decreasing $^1$H polarization $P_H$)." does not make much sense to me.

We thank the editor for raining this point. We have amended this sentence in the manuscript to:

Once the $^{13}$C NMR peak CoG normalized deviation $\delta_{\omega_0}$ falls below zero the $^1$H polarization $P_H$ rapidly drops towards negative values, see the Supplement.

The modified Figure 6 with a best fit of the linear region and negative values of $\delta_{\omega_0}$ does not appear in the ms.

We find the current Figure 5 far more intuitive, and it also provides a much clearer link to the level of $^1$H polarization $P_H$. We included this Figure in the finalized version of the response letter and will keep it as such in the manuscript.

The initials QC should be changed to QS in the Acknowledgements.

This change has been made.